# C16orf72/HAPSTR1/TAPR1 functions with BRCA1/Senataxin to modulate replication-associated R-loops and confer resistance to PARP disruption

Abhishek Bharadwaj Sharma [1,6], Muhammad Khairul Ramlee[1,6], Joel Kosmin[1], Martin R. Higgs [2], Amy Wolstenholme[1], George E. Ronson [1,2], Dylan Jones[3], Daniel Ebner [3], Noor Shamkhi[1], David Sims[4], Paul W. G. Wijnhoven[5], Josep V. Forment[5], Ian Gibbs-Seymour [1] & Nicholas D. Lakin [1] ✉

While the toxicity of PARP inhibitors to cells with defects in homologous recombination (HR) is well established, other synthetic lethal interactions with PARP1/PARP2 disruption are poorly defined. To inform on these mechanisms we conducted a genome-wide screen for genes that are synthetic lethal with *PARP1/2* gene disruption and identified *C16orf72/HAPSTR1/TAPR1* as a novel modulator of replication-associated R-loops. *C16orf72* is critical to facilitate replication fork restart, suppress DNA damage and maintain genome stability in response to replication stress. Importantly, C16orf72 and PARP1/2 function in parallel pathways to suppress DNA:RNA hybrids that accumulate at stalled replication forks. Mechanistically, this is achieved through an interaction of C16orf72 with BRCA1 and the RNA/DNA helicase Senataxin to facilitate their recruitment to RNA:DNA hybrids and confer resistance to PARP inhibitors. Together, this identifies a C16orf72/Senataxin/BRCA1-dependent pathway to suppress replication-associated R-loop accumulation, maintain genome stability and confer resistance to PARP inhibitors.

DNA repair is critical to maintain genome integrity and defects in these pathways result in increased mutagenesis and chromosome instability that contributes towards a variety of pathologies, including malignancy. Whilst the mechanisms that resolve different varieties of DNA damage are becoming increasingly well defined, understanding how these processes integrate to promote DNA damage tolerance if a particular repair pathway fails will provide critical information that identifies genetic vulnerabilities to exploit in the clinic. For example, whilst DNA repair mechanisms are often dysfunctional in cancer cells, compensatory error-prone pathways maintain cell viability to promote genome instability and tumour progression[1]. Understanding these networks will not only provide insights into how cells maintain genome integrity, but also identify genetic vulnerabilities to target in cancer therapies. The founding example of this strategy is inhibition of Poly(ADP-ribose) polymerases (PARPs) to target homologous recombination (HR) deficient cells[2,3] and PARP inhibitors (PARPi) are currently in use to treat HR-defective tumours[4].

[1]Department of Biochemistry, University of Oxford, South Parks Road, Oxford, UK. [2]Institute of Cancer and Genomic Sciences, University of Birmingham, Edgbaston, Birmingham, UK. [3]Target Discovery Institute, Nuffield Department of Medicine, University of Oxford, Oxford, UK. [4]Weatherall Institute of Molecular Medicine, University of Oxford, John Radcliffe Hospital, Oxford, UK. [5]Early Oncology R&D, AstraZeneca, 1 Francis Crick Avenue, Cambridge Biomedical Campus, Cambridge CB2 0AA, UK. [6]These authors contributed equally: Abhishek Bharadwaj Sharma, Muhammad Khairul Ramlee. ✉e-mail: nicholas.lakin@bioch.ox.ac.uk

PARPs catalyse the addition of single or poly-ADP-ribose moieties onto target proteins by mono-ADP-ribosylation or poly-ADP-ribosylation respectively. Whilst ADP-ribosylation (ADPr) has been implicated in a variety of cellular processes, the best defined role of these enzymes is in DNA repair, specifically of DNA strand breaks[5,6]. PARP1 and PARP2 are activated on binding single strand breaks (SSBs) and ADPr substrates to promote assembly of DNA repair and chromatin remodelling factors at DNA lesions[7]. PARP1 also promotes double strand break (DSB) repair by alternative non-homologous end-joining (alt-NHEJ), a pathway that employs micro-homology-based repair mechanisms to resolve lesions in the absence of classic NHEJ (c-NHEJ)[8–10]. More recently, PARP1/2 have been implicated in regulating various aspects of DNA replication including Okazaki fragment processing[11], recruitment of MRE11 to stalled/damaged replication forks[12–14], maintenance of regressed replication forks through inhibition of RECQ1[15], and assembly of HR factors at these structures[16].

Current models for the synthetic lethal interaction between PARPi and HR centre on the disruption of recombination-based repair mechanisms, most notably during DNA replication. For example, PARPi result in the accumulation of unrepaired SSBs, or trap PARP1 and PARP2 at sites of DNA damage. Collision of the DNA replication machinery with these structures causes replication fork stalling and/or collapse that requires HR-mediated repair[4,17]. As a consequence, HR-defective tumours experience elevated levels of PARPi-induced DNA damage that are channelled through alternative mutagenic repair mechanisms, resulting in cell death. More recently, the formation of post-replicative ssDNA gaps was found to correlate with PARPi sensitivity or resistance of HR-defective cells[18–20]. This, taken together with the observations that PARP1/2 are required for a backup pathway to process Okazaki fragments[11,21], has led to an alternative model whereby PARPi toxicity is mediated through an inability to repair DNA gaps to complete maturation of nascent DNA strands during replication[22].

Given the importance of PARPi in the clinic, a variety of screens have been performed to identify novel synthetic lethal interactions with PARPi, in addition to mechanisms by which HR-defective cells become resistant to these agents. The ability of PARPi to trap PARP1 on chromatin is a major contributor to their toxicity in HR-defective cells[17,23,24], underscoring the importance of this approach. However, gene disruption or depletion of PARP1 and/or PARP2 is also toxic in HR-defective cells[2,3,16], indicating the synthetic lethal interaction between PARP1/2 and HR extends beyond PARP-trapping. Given loss of PARP-dependent DNA repair may lead to distinct lesions from those produced by PARP1/2 trapped on chromatin, it is also important to identify synthetic lethal interactions with *PARP1/2* gene disruption. Here we perform such an approach to identify genes that are synthetic lethal in cells deleted for the *PARP1/2* genes. We identify *C16orf72/HAPSTR1/TAPR1* as a gene that is synthetic lethal with *PARP1/2* gene disruption and allows cells to tolerate replication stress. Further, we identify that this is achieved though C16orf72 functioning with BRCA1 and Senataxin to allow replication fork progression by suppressing R-loops generated in response to replication stress and that disruption of this pathway is synthetic lethal with PARP dysfunction.

## Results

### C16orf72 is synthetic lethal with *PARP1/PARP2* gene disruption
Whilst screens have identified genetic vulnerabilities that sensitise cells to PARPi[25,26], to a large extent toxicity to these agents is driven by trapping PARPs at DNA lesions[4,17,23]. Our previous work identified that *PARP1/PARP2* gene disruption is also synthetic lethal with HR dysfunction[16]. Therefore, as an alternative strategy we undertook a genome-wide CRISPR screen to identify genes that are required for survival of *parp1/2Δ* cells. U2OS wild-type and *parp1/2Δ* cells[16] were transduced with the human TKOv3 lentiviral pooled library that contains 70,948 guides targeting 18,053 genes[27]. Following selection and passaging of cells, genomic DNA was extracted and CRISPR guides

present in genomic DNA amplified and sequenced. Guide depletion in *parp1/2Δ* cells relative to parental controls was analysed using the MAGeCK pipeline[28].

We identified ten genes with a false discovery rate (FDR) cut-off of <0.01 whose loss of function significantly compromised the viability of the *parp1/2Δ* cells (Fig. 1a; Supplementary Data 1). Of these genes, we focussed our attention on *C16orf72/HAPSTR1/TAPR1*, a relatively uncharacterised gene that has been implicated in p53 regulation in response to telomere erosion and a coordinator of the cellular stress response[29,30]. Whilst parental cells tolerate siRNA-mediated depletion of *C16orf72*, it is toxic in *parp1/2Δ* cells (Fig. 1b), independently validating the synthetic lethal interaction between *C16orf72* and *PARP1/PARP2*. Although we observe a reduction in cell viability when C16orf72 is depleted in *parp1Δ* cells, this is significantly reduced when *PARP1* and *PARP2* are disrupted in combination, indicating redundancy between PARP1 and PARP2 in maintaining cell viability in the absence of C16orf72 (Fig. 1c). Consistent with this observation, *c16orf72Δ* U2OS cells (Supplementary Fig. 1) display increased sensitivity towards the PARP inhibitor olaparib, and this phenotype is rescued by stable expression of exogenous FLAG-C16orf72 (Fig. 1d and Supplementary Figs. 1b, 2a). Disruption of *C16orf72* was also identified in a screen for synthetic lethal interactions with ATR inhibitors (ATRi)[31]. Accordingly, we find that *c16orf72Δ* cells also display sensitivity to ATRi that is rescued by expression of exogenous FLAG-C16orf72 (Fig. 1e). Both PARPi and ATRi sensitivity is evident in *c16orf72Δ* RPE1 cells, indicating this observation is not limited to U2OS cells (Supplementary Fig. 1c and 2b, c). Together, these data identify *C16orf72* as a novel gene that is synthetic lethal with compromised PARP and/or ATR activities.

### C16orf72 is required for tolerance of cells to replication stress
Given that genes which are synthetic lethal with PARP and ATR inhibitors often have a role in the DNA damage response[25,26,31], we considered whether a critical role for C16orf72 in the cellular stress response is to maintain genome integrity following genotoxic stress. To test this, we exposed *c16orf72Δ* cells to a variety of genotoxins and used clonogenic survival assays to assess whether *C16orf72* is required for cells to combat a particular type of DNA damage. The *c16orf72Δ* cell lines are not overtly sensitive to phleomycin, methyl methanesulfonate (MMS) or mitomycin C that induce DNA strand breaks, DNA base alkylation and DNA crosslinks, respectively (Fig. 2a). However, disruption of *C16orf72* in two independent U2OS clones results in sensitivity of cells to the replication stress induced by hydroxyurea (HU) and aphidicolin (Fig. 2b, c and Supplementary Fig. 3). Expression of *C16orf72* was able to restore the tolerance of *c16orf72Δ* cells to HU and aphidicolin, confirming the dependence of these phenotypes on C16orf72 (Fig. 2b, c and Supplementary Fig. 3). To assess the potential role of C16orf72 in replication stress further, we tested whether it assembles at sites of stalled and/or damaged replication forks. Consistent with this hypothesis, we observe accumulation of C16orf72 in chromatin following exposure of cells to HU (Fig. 2d). Collectively, these data indicate that whilst C16orf72 is not required for cells to tolerate DNA strand breaks or base damage, it is required to maintain cell viability in response replication stress.

### C16orf72 is required to suppress replication-associated DNA damage and promote replication fork recovery
Given that *c16orf72Δ* cells are sensitive to agents that perturb replication dynamics, we considered whether this gene is required to maintain genome stability during DNA replication. Exposure of *c16orf72Δ* cells to HU results in elevated levels of γH2AX (Fig. 3a and Supplementary Fig. 4a), indicating that C16orf72 is required to suppress accumulation of DNA damage in response to replication fork stalling. To assess whether this is due to DSBs formation caused by replication fork catastrophe, we exploited quantitative image-based cytometry (QIBC) to assess DSB formation in response to replication stress by

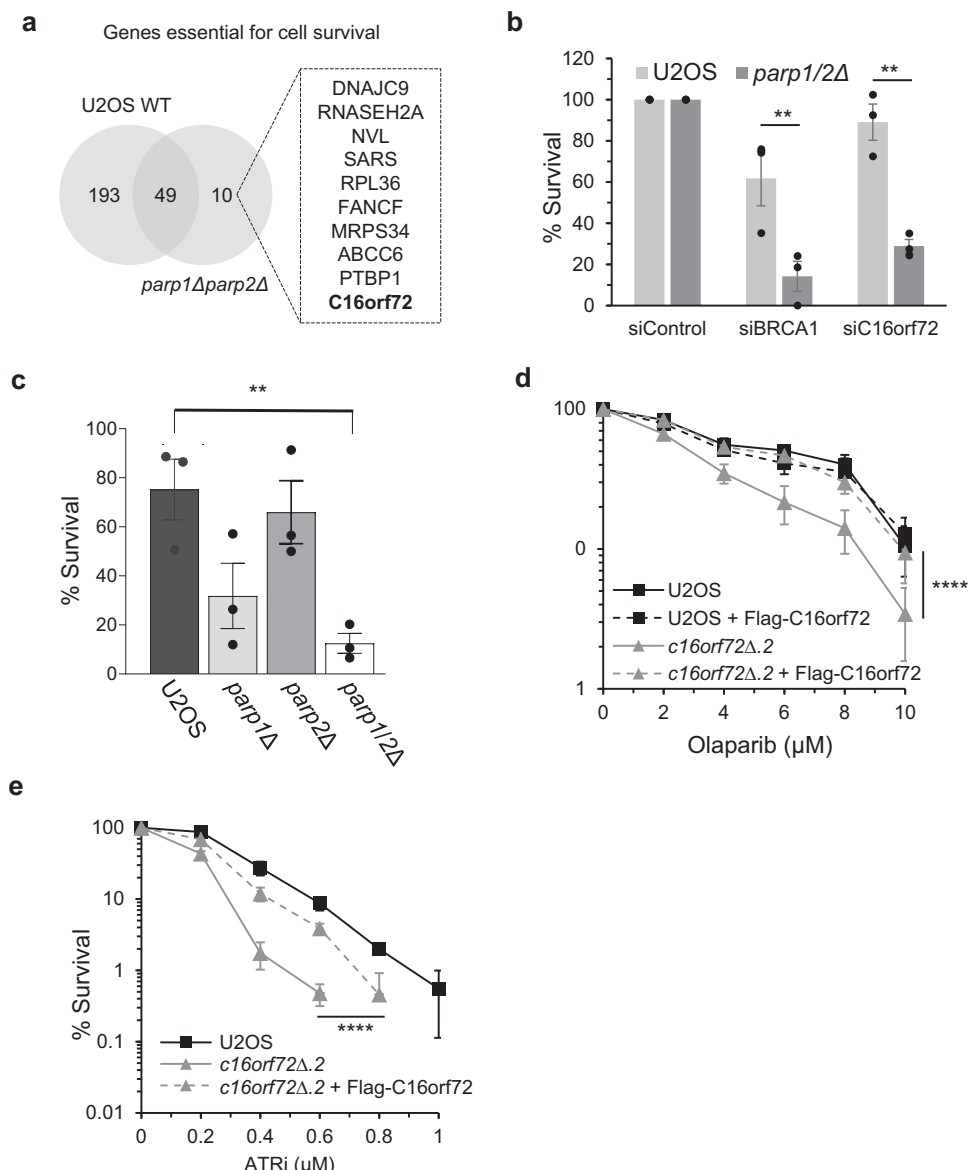

**Fig. 1 | Depletion of *C16orf72* is synthetic lethal with disruption of PARPs. a** A genome-wide CRISPR screen identified *C16orf72* as one of ten genes whose disruption negatively affected the survival of *PARP1/PARP2* double knock-out cells (*parp1Δparp2Δ*) as compared to wild-type U2OS cells. **b** Clonogenic survival assay of wild-type U2OS and *parp1/2Δ* cells treated with siRNA targeting *C16orf72*. Treatment with non-targeting siRNA (siNT) and siRNA targeting *BRCA1* act as negative and positive controls, respectively. **c** Clonogenic survival assay of wild-type U2OS, *PARP1* knock-out (*parp1Δ*), *PARP2* knock-out (*parp2Δ*) and *PARP1/PARP2* double knock-out (*parp1/2Δ*) cells transfected with siC16orf72. **d** Clonogenic survival assay of *C16orf72* knock-out cells (*c16orf72Δ.2*) and complemented cells (*c16orf72Δ.2* cells expressing Flag-C16orf72) treated with increasing concentration of Olaparib for 9 days. **e** Clonogenic survival assay of *C16orf72* knock-out cells (*c16orf72Δ.2*) and complemented cells (*c16orf72Δ.2* + Flag-C16orf72) treated with increasing concentration of ATRi for 9 days. For all clonogenic survival assay plots, mean values ± SEM of three independent biological repeats shown. Statistical analysis was performed using either a one-way ANOVA with a Bonferroni post-hoc analysis (b, c), or two-way ANOVA (d, e); *$p < 0.05$; **$p < 0.01$; ***$p < 0.001$; ****$p < 0.0001$. Source data are provided as a Source Data file.

quantifying γH2AX in cells with elevated RPA[32]. Strikingly, *c16orf72Δ* cells display increased numbers of RPA70/γH2AX double-positive cells relative to parental controls, indicating of an elevated incidence of cells undergoing replication catastrophe (Fig. 3b).

Next, to investigate the role of C16orf72 in replication repair mechanisms, we assessed the ability of cells to recover after a transient exposure to HU. Relative to parental cells, γH2AX levels persist in *c16orf72Δ* cells following removal of HU (Fig. 4a and Supplementary Fig. 4b), suggesting difficulties in the ability of cells to repair and/or restart stalled replication forks. To assess this further, we performed DNA fibre analysis to determine the role of C16orf72 in various aspects of replication dynamics. Disruption of *c16orf72* significantly reduces replication fork speed either in the absence or presence of HU,

indicating difficulties in replication fork progression in cells lacking C16orf72 (Fig. 4b). Further, *c16orf72Δ* cells display elevated levels of stalled replication forks, either in untreated cells or following a transient exposure to HU (Fig. 4c). Both phenotypes are dependent on expression of the C16orf72 protein. However, replication origin firing remains largely unaffected in *c16orf72Δ* (Fig. 4d), suggesting no major role for C16orf72 in controlling the intra-S phase checkpoint. Additionally, C16orf72 status has no impact on cell proliferation, or the distribution of cells at different stages of the cell cycle (Supplementary Fig. 5a, b). Moreover, C16orf72 is not detectable in chromatin prepared form untreated cells (Fig. 2d) and proximity ligation assays reveal it does not co-localise with the replication fork component MCM2 (Supplementary Fig. 5c). Together, these data indicate that C16orf72 is

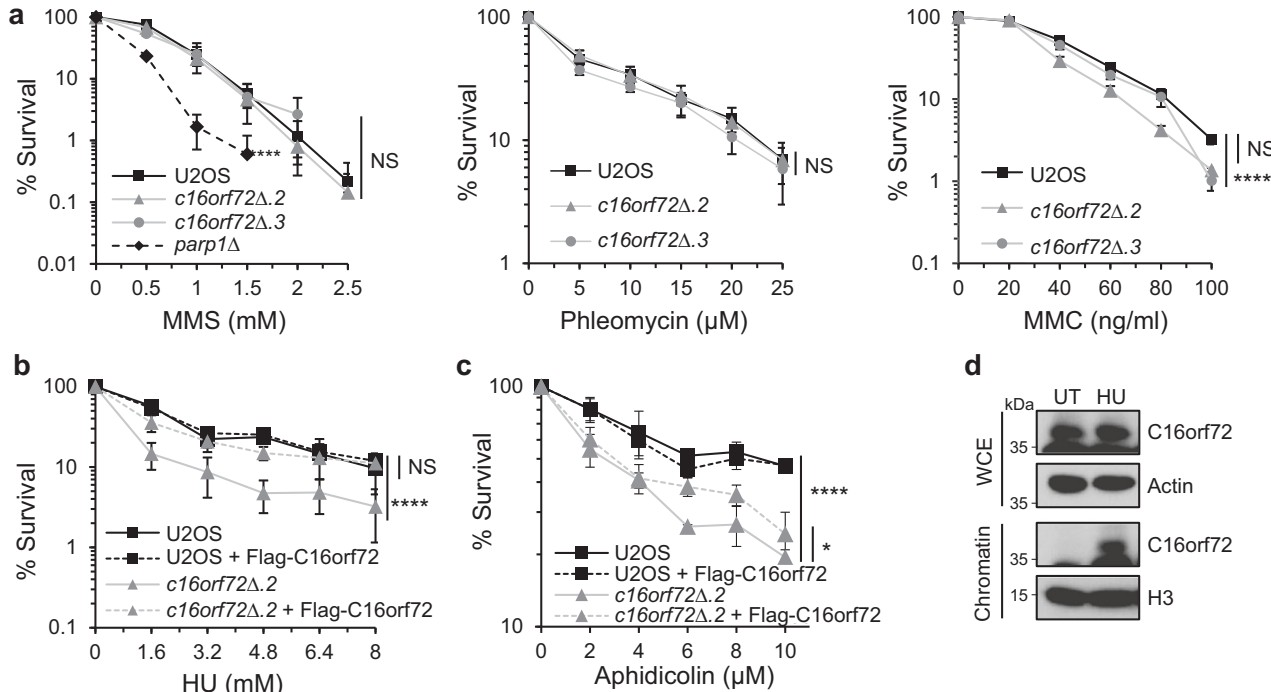

**Fig. 2 | *C16orf72* is required for tolerance against replication stress. a** Clonogenic survival assay of *C16orf72* knock-out cells (*c16orf72Δ.2* and *c16orf72Δ.3*) treated with increasing concentration of methyl methanesulfonate (MMS) for 1 h (left), phleomycin for 1 h (middle) and mitomycin C (MMC) for 24 h (right). *PARP1* knock-out cells (*parp1Δ*) were used as positive control for MMS sensitivity. **b** Clonogenic survival assay of *C16orf72* knock-out cells (*c16orf72Δ.2*) and complemented cells (*c16orf72Δ.2* expressing Flag-C16orf72) treated with increasing concentration of the replication stress-inducing agent, hydroxyurea (HU), for 24 h. **c** Clonogenic survival assay of *C16orf72* knock-out cells (*c16orf72Δ.2*) and complemented cells (*c16orf72Δ.2* + Flag-C16orf72) treated with increasing concentration of the

replication stress-inducing agent aphidicolin, for 24 h. **d** Western blot analysis of C16orf72 protein in whole-cell extract (WCE) and chromatin-enriched fraction of U2OS cells treated with 2 mM hydroxyurea (HU) for 24 h. Extracts were prepared for untreated or HU-treated cells and western blotting performed with the indicated antibodies. Images are representative of 3 biological repeats. For all clonogenic survival assay plots, mean values ± SEM of 3 biological independent experiments are shown. Statistical analysis performed using two-way ANOVA with replication; *$p < 0.05$; **$p < 0.01$; ***$p < 0.001$; ****$p < 0.0001$. Source data are provided as a Source Data file.

not a component of the replication fork but is instead required to process stalled and/or damaged replication forks to promote replication progression. Consistent with this, *c16orf72Δ* cells display elevated levels of 53BP1 bodies in the following G1 phase of the cell cycle, especially after exogenous replication stress (Fig. 4e and Supplementary Fig. 6), suggesting persistence of replication stress into the following cell cycle. In summary, these data indicate a critical requirement for C16orf72 in repair of stalled/damaged replication forks to maintain genome stability in response to replication stress.

## C16orf72 suppresses R-loop formation in response to replication stress

Next, we considered which pathway C16orf72 regulates to maintain cell viability in the absence of PARP1/2. Given PARP inhibition is synthetic lethal with HR and the role of recombination-based repair mechanisms in replication fork recovery[33], we initially considered whether C16orf72 regulates HR in response to replication stress. However, RAD51 nuclear foci are able to form in *c16orf72Δ* cells following HU exposure, and consistent with increased DNA damage in these cells, their frequency is elevated relative to parental control cells, particularly at early timer points (Supplementary Fig. 7a). Further, disruption of *C16orf72* and siRNA depletion of BRCA2 in combination further sensitises cells to olaparib, indicating the two genes function in parallel pathways in terms of synthetic lethality with PARPi (Supplementary Fig. 7b). Taken together, these data indicate that the synthetic lethal interaction between *C16orf72* and *PARP1/2* gene disruption is not a consequence of a role for C16orf72 in BRCA2/RAD51-dependent HR. Therefore, we tested whether other genes known to be synthetic lethal

with PARPi (e.g., FANCD2, BRCA1, RNaseH2A) regulate C16orf72 chromatin recruitment in response to HU. Strikingly, whilst depletion of BRCA1 or FANCD2 had little impact on enrichment of C16orf72 onto chromatin following HU exposure, this was compromised upon depletion of RNaseH2A, indicating C16orf72 may play a role in an RNaseH2A-mediated mechanism(s) in response to replication stress (Fig. 5a).

RNaseH2A is the catalytic subunit of the RNaseH2 complex that excises erroneously incorporated ribonucleotides from duplex DNA in the ribonucleotide excision repair (RER) pathway[34]. Disruption of this function is synthetic lethal with PARPi due to the lesions being channelled through an alternative topoisomerase 1-mediated pathway that leads to PARP-trapping[26]. Therefore, we next asked if the absence of C16orf72 leads to an increase in genomic ribonucleotide incorporation by subjecting the DNA isolated from cells to alkaline hydrolysis and alkaline gel electrophoresis[35]. However, whereas RNaseH2A-depleted cells show an increase in rNTP incorporation, we observe no such affect in *c16orf72Δ* cells (Supplementary Fig. 8a–c), indicating no predominant role for C16orf72 in RER.

RNaseH2A has also been implicated in the resolution of R-loops, transient DNA-RNA hybrids that form when nascent RNA anneals to complementary DNA on the template strand behind the RNA polymerase. These structures can form under a variety of circumstances, including when replication forks collide with transcription complexes, and their deregulation results in elevated DNA damage and genome instability[36]. Therefore, we investigated whether C16orf72 contributes to R-loop homoeostasis. Initially, using proximity ligation assays (PLA) to assess the proximity of proteins in the nucleus, we tested whether

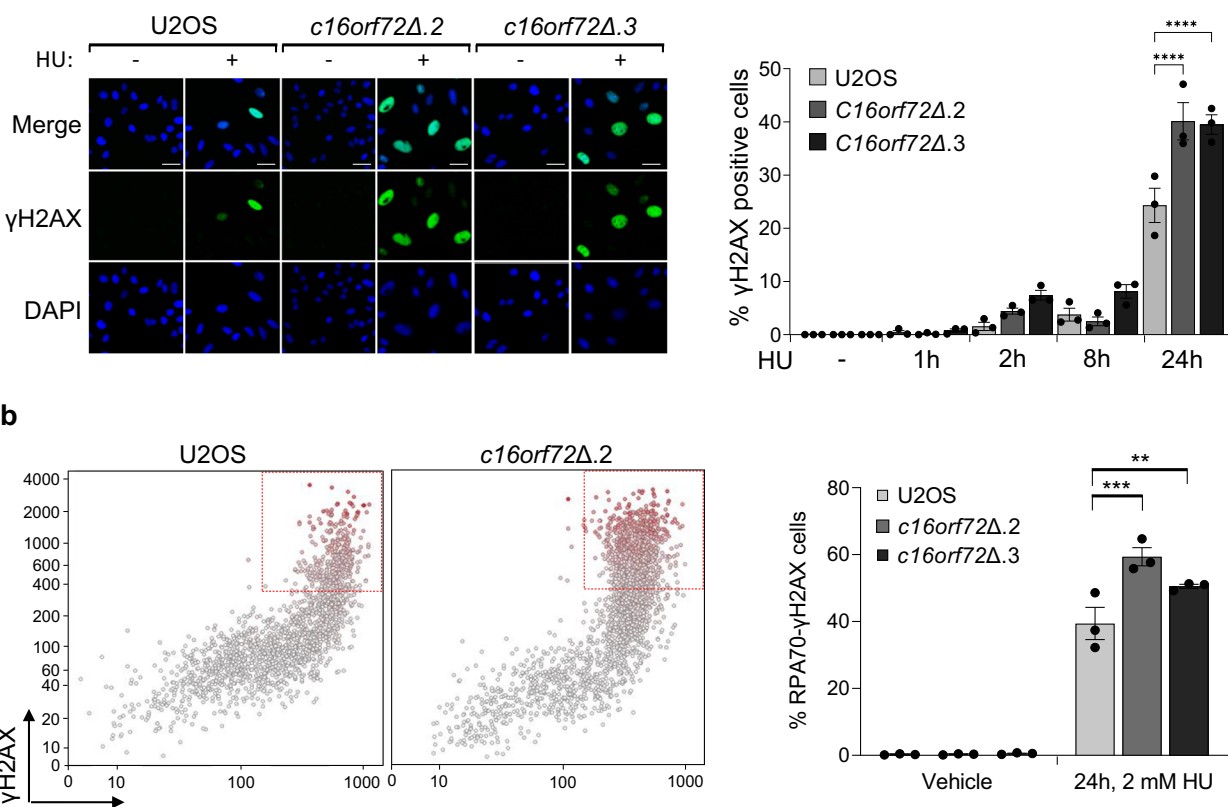

**Fig. 3 | C16orf72-depleted cells display elevated levels of DNA damage.**
**a** Quantitative image-based cytometry (QIBC) of γH2AX levels in wild-type U2OS and C16orf72 knock-out cells (*C16orf72Δ.2* and *C16orf72Δ.3*) treated with 2 mM hydroxyurea (HU) for 0, 1 h, 2 h, 8 h or 24 h. Left: representative images of γH2AX induction in the cells. Scale bar represents 20 μm. Right: Mean % γH2AX positive cells ± SEM of 3 biological independent experiments with at least 500 cells analysed per condition. **b** QIBC co-staining of γH2AX and RPA70 in wild-type U2OS and

*C16orf72* knock-out cells (*c16orf72Δ.2* and *c16orf72Δ.3*) treated with 2 mM hydroxyurea (HU) for 24 h. Left: representative results of elevated γH2AX and RPA70 levels in *c16orf72Δ.2* cells. Right: mean values ± SEM of 3 biological independent experiments. Statistical analysis from at least 3 independent experiments was performed using Ordinary one-way ANOVA with a Bonferroni post-hoc analysis; $**p < 0.01$; $****p < 0.0001$. Source data are provided as a Source Data file.

C16orf72 localises to R-loops using the S9.6 antibody that recognises DNA:RNA hybrids[37]. We observe a >6-fold increase in the PLA signal between S9.6 and C16orf72 in response to HU relative to untreated cells and single antibody controls (S9.6 and C16orf72; Fig. 5b, c), suggesting that C16orf72 and DNA:RNA hybrids are in close proximity in response to replication stress. Next, by staining cells with S9.6, we assessed the impact of C16orf72 loss on nuclear R-loop levels in response to HU. In parental U2OS cells, we observe an induction of pan-nuclear and focal R-loop staining at 2-hour HU-treatment (Fig. 5d). The nuclear S9.6 signal is sensitive to RNaseH treatment, underlining the specificity of the antibody for R-loops (Fig. 5d). Strikingly, *c16orf72Δ* cells exhibit elevated levels of these structures after 2 h of HU treatment, and these persisted at 24 hours post-HU relative to parental U2OS cells (Fig. 5d and Supplementary Fig. 9a–c). Expression of exogenous FLAG-HA-C16orf72 in *c16orf72Δ* cells was able to reduce HU-induced R-loop levels to those observed in control cells, indicating the dependence of this phenotype on C16orf72 (Fig. 5e and Supplementary Fig. 9b). Additionally, treatment of cells with the RNA polymerase II inhibitor 5,6-Dichlorobenzimidazole 1-β-D-ribofuranoside (DRB) supresses the accumulation of HU-induced R-loops to background levels in all cell lines tested (Fig. 5f), indicating the dependence of these structures on active transcription. Taken together, these data identify a role for C16orf72 in suppressing replication stress-induced R-loops to maintain genome stability.

## C16orf72 functions with BRCA1 and Senataxin to supress HU-induced R-loops

Having established that C16orf72 influences R-loop levels in response to replication stress, we sought to further define the pathways it functions in to achieve this regulation. Initially, we tested whether RNaseH2A and C16orf72 act in the same pathway to suppress R-loop formation. Similar to C16orf72 disruption/depletion, we observed a significant increase in HU-induced R-loops in the RNaseH2A-depleted U2OS cells, or RNaseH2A HeLa knock out cells relative to parental control cells (Fig. 6a, b and Supplementary Fig. 9d). Surprisingly, however, in both cell types we observe that disruption of RNaseH2A and C16orf72 in combination is additive in terms of accumulation of HU-induced R-loops (Fig. 6a, b), indicating these genes act in parallel pathways to suppress accumulation of R-loops in response to replication stress.

Given the above, we considered whether C16orf72 functions in another pathway to regulate R-loop accumulation. A variety of factors have been implicated in resolving R-loops that arise at DNA breaks or replication-transcription conflicts. These include the RNA helicases Senataxin, DHX9, DDX5[38–42] and ATAD5 that unloads PCNA to promote RNA helicase recruitment to replication forks[43]. Therefore, we initially tested whether depletion of these factors elevates replication-stress induced R-loops, similar to *C16orf72*. Strikingly, whilst depletion of DDX5, ATAD5 or DHX9 did not further elevate R-loops in response to

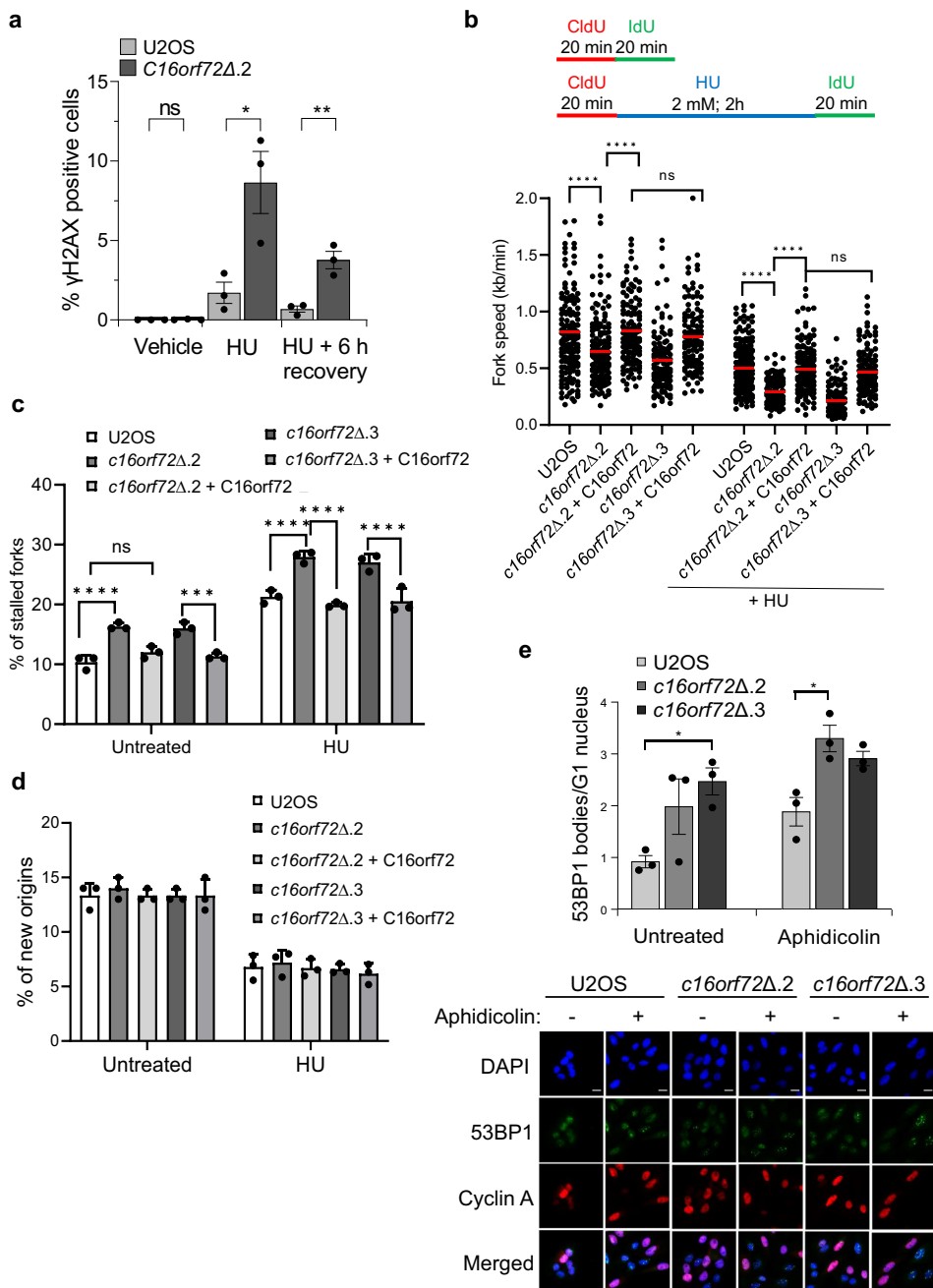

**Fig. 4 | Depletion of *C16orf72* leads to replication stress. a** Immuno-fluorescence microscopy quantification of nuclear γH2AX intensity in wild-type U2OS and *C16orf72* knock-out cells (*c16orf72Δ.2*) treated with 200 µM hydroxyurea (HU) for 24 h with or without 6 h recovery thereafter. Mean % γH2AX positive cells ± SEM of 3 biological independent experiments with at least 1500 cells analysed per condition. **b**–**d** DNA fibre spreading analysis of wild-type U2OS and *C16orf72Δ* cells. U2OS, *C16orf72Δ* (*c16orf72Δ.2* and *c16orf72Δ.3*) and *C16orf72Δ* cells complemented with Flag-C16orf72 (+ C16orf72) were left untreated, or exposed to 2 mM hydroxyurea (HU) for 2 h according to the schematic (upper panel). CldU/IdU labelling patterns and tract length were used to determine replication fork speed (**b**), percentage of DNA fibres with stalled forks (**c**) or new origins (**d**). Data presented are derived from 3 biological independent experiments. Mean values are represented +/− SEM.

**e** Quantitative image-based cytometry (QIBC) of 53BP1 level in wild-type U2OS and *C16orf72* knock-out cells (*c16orf72Δ.2* and *c16orf72Δ.3*) left untreated or exposed to 0.2 µM aphidicolin for 24 h. Below: representative images of 53BP1 induction in the cells. Scale bars represent 20 µm. Above: mean nuclear 53BP1 bodies per G1 nucleus (Cyclin A negative) ± SEM of 3 biological independent experiments with at least 400 G1 cells (cyclin A-negative cells) analysed per condition. For all plots, mean ± SEM of three independent biological repeats shown. Statistical analysis for (**b**) was Mann-Whitney non-parametric. In all other instances statistical significance was determined using a Ordinary one-way ANOVA with a Bonferroni post-hoc analysis. *$p < 0.05$; **$p < 0.01$; ***$p < 0.001$; ****$p < 0.0001$. Source data are provided as a Source Data file.

HU relative to untreated controls (Supplementary Fig. 10), this did occur upon depletion of Senataxin (Fig. 6c). Importantly, Senataxin depletion failed to further increase these structures in *c16orf72Δ* cells (Fig. 6c), indicating C16orf72 and Senataxin function in the same pathway to suppress replication stress-induced R-loops. Given

Senataxin functions alongside BRCA1 to regulate R-loops during DNA repair[44,45], we also tested whether this gene functions with C16orf72 to regulate these structures in response to replications stress. Consistent with this, BRCA1 depletion does not further increase HU-induced R-loops in *c16orf72Δ* cells indicating it is also epistatic with C16orf72 to

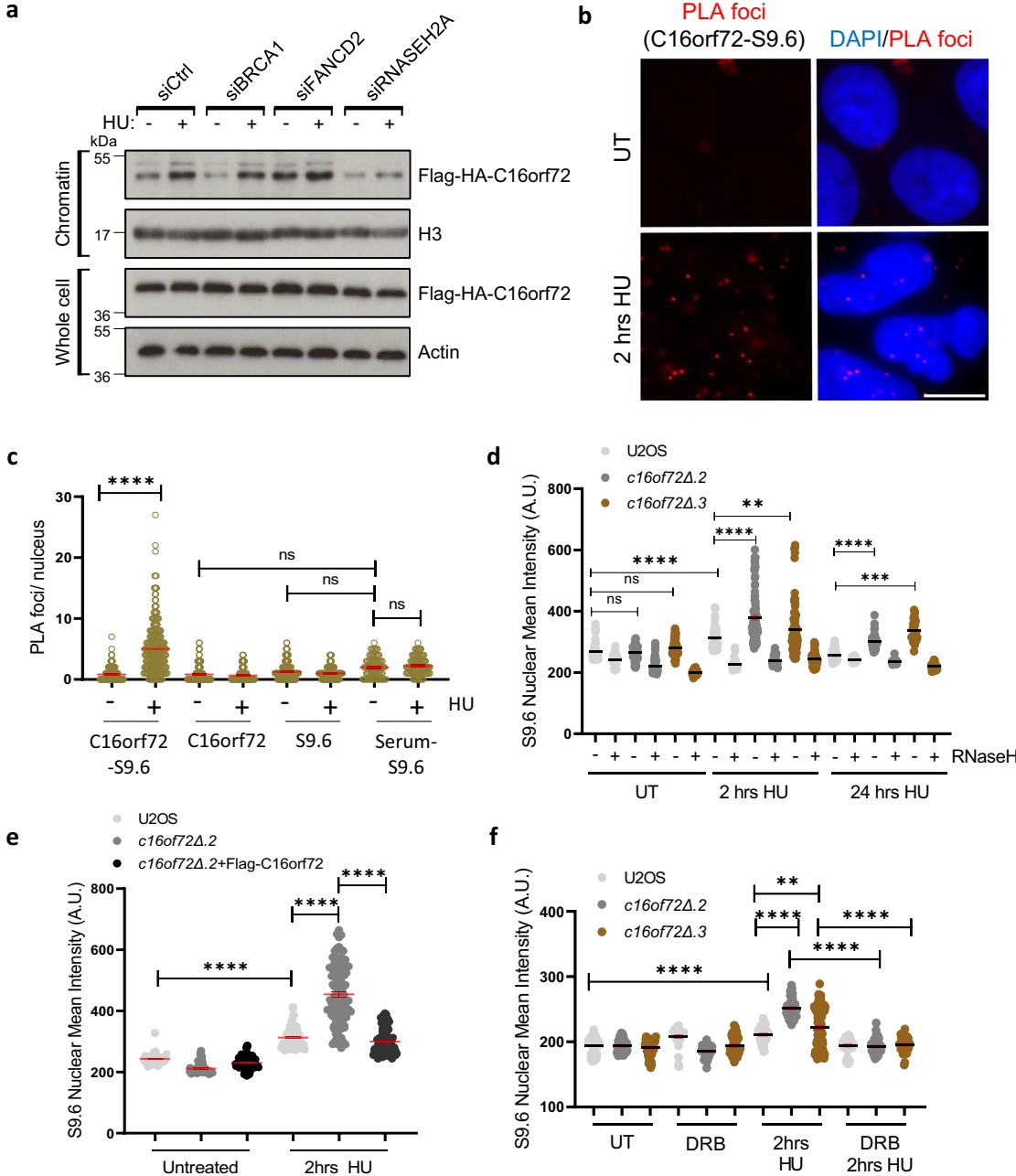

**Fig. 5 | C16orf72 modulates R-loop homoeostasis in response to replication stress. a** Cells expressing Flag-HA-tagged C16orf72 were transfected with control siRNA, or siRNA targeting *BRCA1*, *FANCD2* or *RNASEH2A*. Cells were left untreated or exposed to 2 mM HU for 24 h prior to preparing cell extracts and western blotting with antibodies as indicated. Images are representative of 2 biological repeats. **b**, **c** U2OS cells were treated with 2 mM HU for 2 hours and subjected to proximity ligation assays using anti-C16orf72 and anti-S9.6 antibodies. Representative images of PLA foci (red) and nuclei (DAPI) are illustrated (**b**). Scale bars represent 10 μm. Graphs (**c**) represent the average foci/nucleus from (**b**). Mean values (red lines) are represented +/− SEM for at least 93 cells examined per treatment over 3 biological independent experiments. **d** Parental U2OS and *c16orf72Δ* cells were treated with HU for the indicated times prior to immuno-fluorescence with the S9.6 antibody. As a control for S9.6 antibody specificity, cells were left untreated or incubated with RNaseH before immuno-fluorescence. Data represents the quantification of nuclear mean intensity of the S9.6 signal. Data presented represent at least 124 cells over 3 (U2OS UT+RNaseH; c16orf72Δ.2 UT+RNaseH; c16orf72Δ.3 2 hrs HU; c16orf72Δ.2 24hrs HU) or 205 cells over 4 (U2OS UT; c16orf72Δ.2 UT; c16orf72Δ.3 UT; U2OS 2 hrs HU; c16orf72Δ.2 2 hrs HU) biological independent experiments. All other samples represent at least 50 cells examined per condition over 2 biological

independent experiments. Mean values are represented (black lines) +/− SEM. **e** Parental U2OS, *c16orf72Δ* cells, or *c16orf72Δ* cells expressing Flag-C16orf72 (*c16orf72Δ* + Flag-C16orf72) were treated with HU for the indicated time prior to immuno-fluorescence with the S9.6 antibody. Data represent the quantification of nuclear mean intensity of the S9.6 signal. Data presented represent at least 150 cells examined per condition over 3 (U2OS UT; c16orf72Δ.2+Flag-C16orf72 2 hrs HU), 4 (c16orf72Δ.2 UT; c16orf72Δ.2+Flag-C16orf72 UT; U2OS 2 hrs HU) or 5 (c16orf72Δ.2 2 hrs HU) biological independent experiments. Mean values are represented (red lines) +/− SEM. **f** The indicated cells were treated with 100 μM DRB for 3 hours, followed by HU in the presence or absence of DRB, as indicated. Cells were subjected to immuno-fluorescence using the S9.6 antibody. Data represent the quantification of nuclear mean intensity of the S9.6 signal. Data presented represent at least 141 cells examined per condition over 3 biological independent experiments with the erxception of U2OS UT; c16orf72Δ.2 UT; U2OS 2 hrs HU; c16orf72Δ.3 2 hrs HU; U2OS DRB + HU; c16orf72Δ.3 DRB + HU (*n* = 4). Mean values are represented (black lines) +/− SEM. All Statistical significance in plots was assessed by Ordinary one-way ANOVA or Kruskal Wallis non-parametric tests (**$*p \leq 0.01$, ***$p \leq 0.001$, ****$p \leq 0.0001$ and ns = not significant). Source data are provided as a Source Data file.

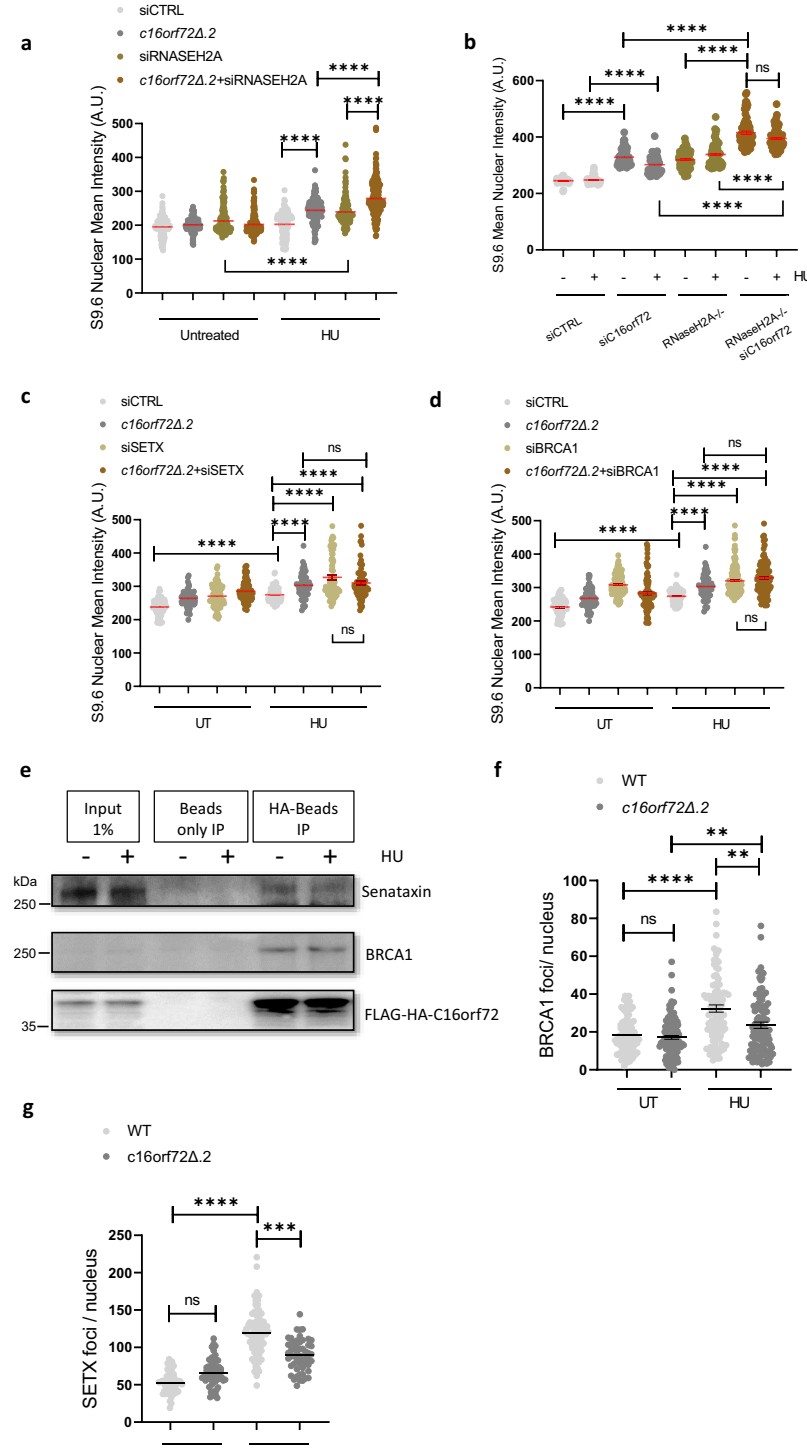

suppress R-loops accumulation at sites of stalled and/or damaged replication forks (Fig. 6d). To further validate this relationship, we tested whether BRCA1 and Senataxin physically interact with C16orf72. Both Senataxin and BRCA1 co-immunoprecipitate with HA-Flag-C16orf72 indicating they form a complex in cells, although this interaction is independent of replication stress (Fig. 6e). C16orf72 is also present in Senataxin immunoprecipitations, validating the interaction between these two proteins (Supplementary Fig. 11a). Previously, the BRCA1-Senataxin interaction was reported to facilitate resolution of R-loops at transcription termination sites. To establish whether C16orf72 assists with this in the context of replication stress, we

assessed whether it is required for BRCA1-Senataxin assembly onto chromatin in response to HU. Whilst HU induced a >1.7-fold increase in BRCA1 nuclear foci in the parental U2OS, c16orf72Δ cells exhibited significant reduction in these structures (Fig. 6f and Supplementary Fig. 11c). Similarly, whilst we observe a >2.7-fold increase in Senataxin foci in response to HU, similar to BRCA1 this is significantly reduced in c16orf72Δ cells (Fig. 6g and Supplementary Fig. 11b). Taken together, these data suggest C16orf72 interacts with both Senataxin and BRCA1 and is required to assemble this complex at sites of replication stress to facilitate R-loop resolution.

**Fig. 6 | C16orf72 and BRCA1/Senataxin interact to modulate R-loop homoeostasis in response to replication stress. a** Parental U2OS or *c16orf72Δ* cells were transfected with either control (siCTRL) or RNaseH2A (siRNaseH2A) siRNA as indicated. Cells were left untreated or exposed to 2 mM HU for 2 hours prior to immuno-fluorescence with the S9.6 antibody. Data represents the quantification of nuclear mean intensity of the S9.6 signal. Mean values (red lines) are represented +/− SEM where at least 138 cells were examined per treatment over 3 biological independent experiments. **b** Parental HeLa cells or RNaseH2 knock-out cells (RNaseH2A-/-) were transfected with either control or C16orf72 siRNA (siCTRL and siC16orf72, respectively) as indicated. Cells were left untreated or exposed to 2 mM HU for 2 hours prior to immuno-fluorescence with the S9.6 antibody. Data represents the quantification of nuclear mean intensity of the S9.6 signal. Mean values (red lines) are represented +/− SEM where at least 210 cells were examined per treatment over 4 biological independent experiments with the exception of RNaseH2A$^{-/-}$/siC16orf72 cells (n = 3). **c** Parental U2OS and *c16orf72Δ* cells were transfected with control (siCTRL) or Senataxin (siSETX) siRNA as indicated. Cells were left untreated or exposed to 2 mM HU for 2 hours, prior to immuno-fluorescence with the S9.6 antibody. Data represent the quantification of nuclear mean intensity of the S9.6 signal. Mean values (red lines) are represented +/− SEM where at least 226 cells were examined per treatment over 4 biological independent

experiments. **d** Parental U2OS and *c16orf72Δ* cells were transfected with control (siCTRL) or BRCA1 (siBRCA1) siRNA as indicated. Cells were left untreated or exposed to 2 mM HU for 2 hours, prior to immuno-fluorescence with the S9.6 antibody. Data represent the quantification of nuclear mean intensity of the S9.6 signal. Mean values (red lines) are represented +/− SEM where at least 234 cells were examined per treatment over 4 biological independent experiments with the exception of c16orf72Δ.2+siBRCA1 HU cells (n = 3). **e** The *c16orf72Δ* cell line expressing c16orf72-HA-Flag were left untreated or exposed to 2 mM HU for 2 hours as indicted. Following whole-cell extract preparation, c16orf72-HA-Flag was immunoprecipitated using anti-HA beads. Inputs or immunoprecipitates were subjected to western blotting using the indicated antibodies. Data are representative of 3 independent experiments. **f, g** Parental and *c16orf72Δ* U2OS cells were treated with 2 mM HU for 2 hours and nuclear foci detected by immunofluorescence using anti-BRCA1 (**f**) or anti-Senataxin (**g**) antibodies. Data represent the mean BRCA1 and SETX foci count/nucleus, respectively. Mean values (black lines) are represented +/− SEM where at least 187 (**f**) or 155 (**g**) cells were examined per treatment over 3 biological independent experiments. All the Statistical significance was assessed by Ordinary one-way ANOVA or Kruskal Wallis non-parametric tests (**$p \leq 0.01$, ***$p \leq 0.001$, ****$p \leq 0.0001$ and ns = not significant). Source data are provided as a Source Data file.

## The C16orf72/BRCA1/Senataxin pathway is required for replication fork recovery in response to replication stress

Given the requirement for C16orf72 in recruiting Senataxin/BRCA1 to sites of replication stress, we next considered the functional significance of this pathway in allowing cells to facilitate replication fork recovery and tolerate replication stress. Consistent with previous observations (Fig. 4c) we observe that replication fork restart is compromised in *c16orf72Δ* cells following a transient exposure to HU (Fig. 7a). Whilst depletion of either Senataxin or BRCA1 result in a similar inability to restart replications forks in parental U2OS cells, it does not further exacerbate this phenotype in *c16orf72Δ* cells (Fig. 7a), indicating that defective restart of replication forks in the absence of Senataxin/BRCA1 or C16orf72 arise through dysfunction of a shared mechanism. Additionally, whilst depletion of Senataxin or BRCA1 sensitises U2OS cells to HU, it does not exacerbate the sensitivity of *c16orf72Δ* cells to this agent (Fig. 7b, c), again indicating Senataxin/BRCA1 and C16orf72 function in the same pathway to allow cells to process and tolerate replication stress.

Recently, PARP1 has been implicated in regulating R-loops and PARPi elevate the levels of these structures[46,47]. Given C16orf72 also regulates R-loop homoeostasis and is required for cell viability in the absence of PARP1/2, we tested whether C16orf72/BRCA1/Senataxin function in a parallel pathway to PARP1/2 to modulate R-loops and whether this may contribute towards the synthetic lethal interaction between these genes. Consistent with previous reports[46,47], we observe that PARPi induce elevated levels of R-loops in U2OS cells (Fig. 7d). Disruption of *c16orf72* further elevates S9.6 staining in cells exposed to PARPi (Fig. 7d), indicating that the elevated levels of R-loops generated upon PARP1/2 disruption are resolved through a C16orf72-dependent pathway. Moreover, we observe that disruption of *C16orf72* does not further exacerbate the sensitivity of BRCA1 depletion to PARPi (Fig. 7e). Additionally, whilst Senataxin sensitises cells to PARPi, similar to BRCA1, depletion of this gene in *c16orf72Δ* cells does not further sensitise them to PARPi (Fig. 7f). In summary, these data indicate C16orf72 works together with BRCA1 and Senataxin in an alternative pathway to PARP1/2 to regulate R-loop levels to maintain cell viability.

## Discussion

The synthetic lethal interaction between PARP inhibition and HR dysfunction is well established and PARPi are being used to treat HR-defective breast and ovarian tumours[4]. However, despite this relationship being established nearly two decades ago, there are several models for the mechanistic basis of this interaction. In addition to being competitive inhibitors that bind to the active site of

PARP1/PARP2, PARPi also stabilises PARP-DNA complexes at DNA lesions. This 'trapping' of PARPs at DNA breaks is a significant contributor to their toxicity in HR-defective cells through causing replication conflicts[17,23] and/or interfering with post-replicative gap-filling and nascent DNA maturation[18–21]. As such, screens have focussed on identifying genes that are toxic to cells in combination with PARPi to extend the repertoire of synthetic lethal interactions beyond HR[26]. However, *PARP1/2* gene deletion is also synthetic lethal with HR[2,3,16], indicating that toxicity of PARP1/2 dysfunction extends beyond PARP-trapping. Here, we adopt an alternative screening approach to identify synthetic lethal interactions with PARP1/2 gene disruption. This identifies several genes whose disruption selectively affects the viability of *parp1/2Δ* cells (Fig. 1a), indicating that trapping of PARP is not necessarily the sole factor in eliciting synergistic lethality with certain genetic perturbations.

Our screen identified *C16orf72/HAPSTR1/TAPR1*, a gene not previously implicated in survival of cells with compromised PARP1/2 function. Similar to defects in HR[16], the major determinant of synthetic lethality with *C16orf72* is PARP1 (Fig. 1c). Consistent with C16orf72 being identified in a synthetic lethal screen with ATR inhibitors (ATRi)[31], we also identify that *c16orf72Δ* cells require ATR catalytic activity for survival (Fig. 1e). A unifying theme of genes that are required for cells to tolerate PARPi and ATRi is that they function in the DDR, most notably in pathways that combat replication stress. Consistent with this, whilst *c16orf72Δ* cells are able to tolerate agents that induce DNA base damage and DSBs, they are sensitive to agents that induce replications stress (Fig. 2). This is reflected in C16orf72 being required to promote replication fork restart, suppress DNA damage, and maintain genome stability in response replication fork stalling/collapse (Figs. 3 and 4).

Recently, C16orf72 has been implicated in suppressing p53 activation and maintenance of cell fitness in response to telomerase inhibition[30]. Our data indicate a broader role for C16orf72 in replication fork recovery mechanisms by resolving R-loops. Given R-loops can form at telomeres[36], it is interesting to speculate that this may contribute to C16orf72's role at these regions of the genome. In a broader context, along with the E3 ubiquitin ligase HUWE1 as a co-factor, C16orf72 has been proposed as a molecular rheostat crucial for cell survival in multiple stress conditions including metabolic, oxidative, hypoxia and genotoxic stresses[29]. Our data indicate loss of C16orf72 does not sensitise cells to genotoxins more generally, including those that induce base damage and DNA strand breaks (Fig. 2). This supports a more specialised role for C16orf72 in DNA replication, rather than the stress response to genomic insults more generally. However, given the

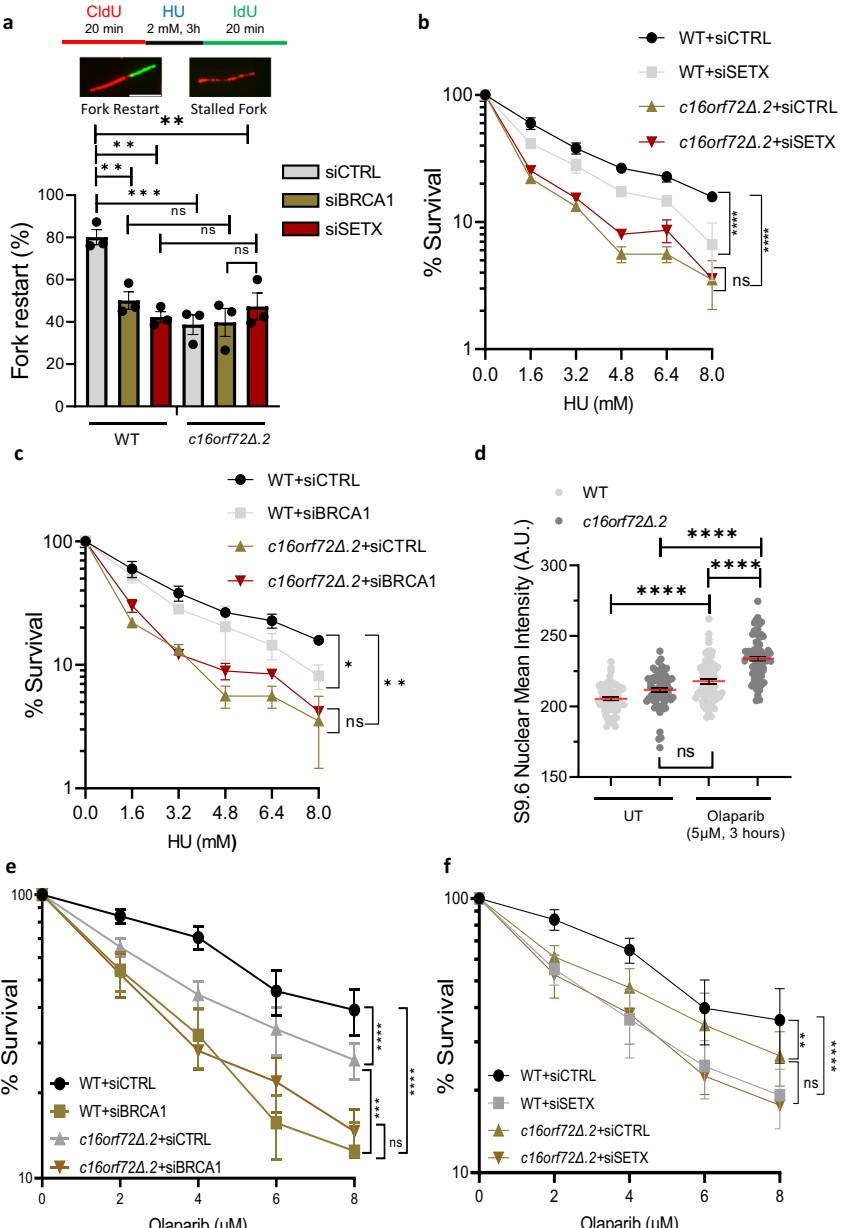

**Fig. 7 | C16orf72 and BRCA1/Senataxin function together to promote replication fork restart and tolerance to replication stress. a** Parental and *c16orf72Δ* U2OS cells were transfected with control (siCTRL), BRCA1, or Senataxin (SETX) siRNA. Following treatment of cells with CldU, HU and IdU as illustrated (top panel), DNA fibres were prepared from cells and quantified for the CldU/IdU staining. 300-900 DNA fibres were analysed per condition from *n* = 3. Fork restart represents the % of fibres that display both CldU and IdU staining (upper panel). Data are presented as mean values +/− SEM. **b, c** Clonogenic survival assay of parental and *c16orf72Δ* U2OS cells transfected with control (siCTRL), Senataxin (siSETX; **b**) or BRCA1 (siBRCA1; **c**) siRNA following exposure to increasing concentrations of HU. Data represent 2 biological repeats. Data are presented as mean values +/− SEM. **d** Parental and *c16orf72Δ* U2OS cells treated with DMSO or Olaparib as indicated

prior to immuno-fluorescence with the S9.6 antibody. Data represents the quantification of nuclear mean intensity of the S9.6 signal. Mean values (red lines) are represented +/− SEM where at least 173 cells were examined per treatment over 3 biological independent experiments. **e, f** Clonogenic survival assay of parental (WT) and *c16orf72Δ* U2OS cells transfected with control (siCTRL), BRCA1 (siBRCA1; **e**) or Senataxin (siSETX; **f**) siRNA following exposure to increasing concentrations of olaparib. Data are presented as mean values +/− SEM where *n* = 5 independent biological repeats for (**e**) and *n* = 3 independent biological repeats for (**f**). In the case of DNA fibre and S9.6 staining, statistical significance was assessed by Ordinary one-way ANOVA or Kruskal Wallis non-parametric test. Clonogenic survival assays were assessed by two-way ANOVA (*$p \le 0.05$, **$p \le 0.01$, ***$p \le 0.001$, ****$p \le 0.0001$ and ns = not significant). Source data are provided as a Source Data file.

role of C16orf72 in R-loop modulation (see below), it will be interesting to assess whether it contributes towards a wider role of this protein in the transcriptional response to multiple stress pathways.

What, then, is the pathway(s) that C16orf72 regulates to allow cells to tolerate replication stress? Given that the synthetic lethal interaction with PARP1/2, the most parsimonious explanation would be a role in HR-mediated repair of stalled/damaged replication forks. However,

in the absence of C16orf72 we observe increased frequency of HU-induced RAD51 foci (Supplementary Fig. 7a), reflecting elevated DNA damage and that initiation of the HR pathway remains intact in these cells. Moreover, we observe that disruption of BRCA2 and C16orf72 in combination is additive in terms of PARPi sensitivity (Supplementary Fig. 7b), further supporting the notion that C16orf72 functions independently of the canonical HR pathway. Instead, we observe that HU-

induced recruitment of C16orf72 to chromatin is reduced in the absence of RNaseH2A, a gene that regulates RER and is synthetic lethal with PARPi[26]. However, we did not observe a significant difference in genome rNMP levels in *c16orf72Δ* cells indicating an intact RER pathway (Supplementary Fig. 8). Rather, these cells displayed a robust and reversible increase in R-loop structures (Fig. 5), indicating a role for C16orf72 in R-loop maintenance to promote viability of cells exposed to PARPi, or undergoing replication stress.

R-loops can form under a variety of circumstances when RNA polymerases pause. As such, they are key determinants of gene expression, particularly with reference to polymerase pausing at transcriptional start and/or termination sites. However, another significant source of R-loops arises from the collision of replication forks with transcription complexes either co-directionally, or head on, and deregulation of these processes results in genome instability[42]. Our data identify C16orf72 as a key determinant of R-loop homoeostasis in response to replication fork stalling. Our initial observations that RNaseH2A is required to recruit C16orf72 to chromatin in response to replication stress suggests a link between this protein and R-loop resolution through RNaseH. However, the elevated levels of R-loops observed in *c16orf72Δ* cells are additive with depletion or disruption of RNAseH2A, indicating a redundant role with RNaseH in processing R-loops. Therefore, we searched for collaborative protein partners of C16orf72 and discovered that Senataxin and BRCA1 are epistatic with this gene in terms of supressing R-loops in response to replication stress and that they physically interact under both the normal and HU-treated conditions (Fig. 6).

Interestingly, Senataxin is a DNA:RNA helicase reported to resolve R-loops not only at transcription pause sites[43,44], but also in response to several stresses including hypoxia, replication stress and replication-transcription conflicts[39,48,49]. The physical and genetic interaction of Senataxin/BRCA1 with C16orf72, and its impact on regulating their localization under the replication stress, sheds light upon how it may resolve the R-loops. Our data support a wider role of this complex in R-loop homoeostasis during replication stress and that together they drive efficient replication fork progression and restart (Fig. 6c, d; Fig. 7a). This is reminiscent of the situation in yeast, where Senataxin associates with replication forks and is required for resolution of replication/transcription conflicts[38]. Interestingly, whilst a number of studies have focussed on active replication forks colliding with ongoing transcription complexes, replication stress can lead to R-loop stabilisation driven by replication-transcription conflicts[50]. Our data implicate C16orf72/BRCA1/Senataxin in R-loop resolution of these structures generated during replication fork stalling. This implies they are important in resolving R-loops either as a consequence of replication forks stalling/pausing in the vicinity of structures that favour R-loop formation, or the collision of RNA polymerases with stalled forks. Whatever the structures, our data clearly point to C16orf72/BRCA1/Senataxin being required for R-loop resolution to restart the replication forks and tolerate replication stress (Fig. 7a–c). In addition to forming in response to polymerase pausing, de novo synthesis of RNA to form R-loops has been implicated in directly regulating the repair process, including transcription-coupled HR[40,44,51–53]. It is interesting to speculate, therefore, that a similar mechanism might be employed through C16orf72/BRCA1/Senataxin in response to replication stress.

Emerging evidence points to R-loops as effective biomarkers for tumour progression and targeted cancer therapies[54]. For example, breast cancer cells with inherited BRCA1 mutations show BRCA1 bound at gene termination sites that would normally engage to resolve R-loops[44]. Additionally, BRCA1 knock-down in ERα+ luminal breast cancer cells increases R-loops[55] and altered C16orf72 expression in breast cancers correlates with decreased survival rates (https://www.cbioportal.org/). Our data indicating a requirement for C16orf72 in recruitment of BRCA1/Senataxin to suppress R-loop accumulation

provides a mechanistic explanation for these observations. Interestingly, BRCA2 and RAD51 mutations have been associated with R-loop accumulation at genes transcribed in early S-phase, with the resulting DNA damage being repaired in mitosis[56,57]. It remains to be tested whether elevated levels of R-loops observed in C16orf72/BRCA1/Senataxin-defective cells similarly occur at genes transcribed in early S-phase. Importantly, however, we observe that PARPi increase R-loops in the absence of HU, and that these structures are further elevated by disruption of *c16orf72* (Fig. 7d). It is interesting to speculate, therefore, that PARP1/2 disruption results in the generation of R-loops that are channelled through a c16orf72/BRCA1/Senataxin-dependent pathway. Our observations that this occurs in the absence of exogenously applied replication stress may provide a mechanistic explanation for the synthetic lethal interaction between C16orf72, BRCA1 and Senataxin with PARPi. Our data also indicate this interaction has a wider application with other DDR inhibitors. For example, ATR signals replication/transcription conflicts to initiate fork restart through Mus81/Rad52/PolD3-dependent mechanisms[58,59]. Our data support a role for C16orf72/BRCA1/Senataxin in a parallel pathway to ATR in order to process R-loops and facilitate replication fork recovery. In addition to providing an explanation for the critical requirement of C16orf72 in cell viability when ATR activity is compromised (Fig. 1e)[31], these observations also suggest ATRi may provide an effective treatment in tumours with elevated R-loops.

## Methods

### Reagents

All gene-targeting siRNAs used in this study were ON-TARGETplus human SMARTPool from Dharmacon. Non-targeting siRNA pool (D-001810-10-05, Dharmacon) was used as control for all siRNA-mediated knock-down experiments. Cells were transfected with 50 nM siRNA twice at 24-hour intervals using Dharmafect-1 (Dharmacon).

The following antibodies were used in the western blot, immunoprecipitation, immuno-fluorescence, and proximity ligation assays described below: HA (Cell Signalling Technology, 3724) at 1:2000, BRCA1 (D-9, SC6954, Santa Cruz Biotechnology) at 1:1000, BRCA1 (07-434, Sigma-Aldrich) at 1:1000, BRCA2 (clone 5.23, 05-666 Millipore) at 1:1000, FANCD2 (Santa Cruz Biotechnology, sc-20022) at 1:500, RNASEH2A(Santa Cruz Biotechnology, sc-515475) at 1:2000, γH2AX for western blot (S319; Abcam, ab11174) at 1:2000, γH2AX for QIBC (Bio-legend, 613402) at 1:500, β-actin (Santa Cruz Biotechnology, sc-1615) at 1:10000, H3 (Abcam, ab12079) at 1:500, phospho-RPA32 S4/S8 (Bethyl, A300-245A) at 1:1000, total RPA32 (Bethyl, A300-244A) at 1:500, RPA70 (Abcam, ab79398) at 1:1000, phospho-DNA-PKcs S2056 (Abcam, ab124918) at 1:2000, total DNA-PKcs (Abcam, ab32566) at 1:1000, phospho-Chk1 S317 (Cell Signalling, D12H3) at 1:1000, total Chk1 (Cell Signalling, 2G1D5) at 1:1000, p53 (Santa Cruz, sc-126-HRP) at 1:500, vinculin (Santa Cruz, sc-73614-HRP) at 1:1000, 53BP1 (Novus Biologicals, NB100-305) at 1:500, cyclin A (Santa Cruz, sc-751) at 1:250, RAD51 (sc-8349, Santa Cruz Biotechnology) at 1:1000, Anti DNA-RNA Hybrid S9.6 clone (MABE1095, Millipore) at 1:100, Senataxin (QQ-7,sc-100319, Santa Cruz Biotechnology) at 1:1000, MCM2 (Abnova, 805-904) at 1:1000 and PCNA (D3H8P XP, Cell signalling) at 1:1000. The polyclonal antibody against the protein encoded by *C16orf72* was raised in rabbit using a 16-residue peptide and was generated by Eurogentec, used at 1:1000.

Olaparib, ATRi (AZD6738) and ATMi (AZD0156) were obtained from Selleck Chemicals or directly from AstraZaneca. MMS, phleomycin, MMC, hydroxyurea, aphidicolin and cycloheximide were obtained from Sigma-Aldrich and MG-132 was obtained from Selleck Chemicals.

### Cell culture

HeLa wild-type and RNASEH2A-null cells were kind gifts from Andrew Jackson[60]. C16orf72 knock-out and overexpressing cells were

generated as described below. All cells were maintained in DMEM (Gibco, 21969-035) supplemented with 10% foetal bovine serum (FBS; Sigma-Aldrich, F7524), L-glutamine (Sigma-Aldrich, G7513) and penicillin-streptomycin (Sigma-Aldrich, P0781). To obtain single-cell suspension, attached cells were treated with trypsin-EDTA solution (Sigma-Aldrich, T3924) for 5–10 min at 37 °C.

### Genome-wide CRISPR screen
U2OS wild-type and *parp1/2Δ* double knock-out cells[16] were transduced with the human TKOvs3 lentiviral pooled library[61] at an MOI of <0.3 and a coverage of 500 cells per guide. The infected cells were selected with puromycin (2 µg/ml) for 72 hours and samples of the resulting selected cell populations collected for DNA extraction (termed "reference" population). The remaining cells were sub-cultured for 12 passages, maintaining a representation of at least 500X (termed "depleted" population) prior to DNA extraction. Genomic DNA was extracted from cells in the reference and depleted cell populations and CRISPR guide RNA cassettes present in gDNA PCR-amplified and subjected to next generation sequencing. Results were analysed using MAGeCK 0.5.9[28] and genes with false discovery rate (FDR) < 0.01 were identified as hits.

### Clonogenic survival assay
Cells were trypsinised, counted and plated at a density of 400–1000 cells per well of 6-well plates and incubated overnight at 37 °C and 5% carbon dioxide level. The next day, cells were treated with drugs, if necessary, for the stipulated duration of time. Thereafter, cells were maintained in normal growth medium until individual colonies were visible to the naked eye, typically 12–15 days after plating the cells. To visualise the colonies, cells were fixed with ice-cold methanol and incubated at −20 °C for 20 min and then stained with crystal violet for 20 min at room temperature. After washing and drying the plates, colonies with more than 50 cells were counted, and relative cell survival was calculated as a percentage with respect to untreated cells.

### Generation of C16orf72 knock-out and complemented cell lines
C16orf72 knock-out clones in U2OS background were generated by excising a part of the coding sequence downstream of the start codon (ATG) in exon 1 using two CRISPR/Cas9 targets [5′-GAGGCCGAGA TCCAGGAGCA(CGG)−3′ + 5′-GTGCCTGGCCGAGGCCGAAC(AGG)−3′ or 5′-GAGGCCGAGATCCAGGAGCA(CGG)−3′ + 5′-CCAGAACTCGGCCA CCGCCG(TGG)−3′]. CRISPR/Cas9 expression plasmids were generated by cloning duplexed oligonucleotides into pSpCas9(BB)−2A-Puro V2.0 (PX459) vector backbone as described previously[62]. U2OS cells were transfected overnight with the two CRISPR/Cas9 plasmids using Lipofectamine 2000 Transfection Reagent (Invitrogen, #11668019) and thereafter selected with 2 µg/ml puromycin for 24 h. Surviving cells were trypsinised and re-plated on 10-cm dishes at densities of 100-500 cells per dish. Cells were incubated at 37 °C and 5% carbon dioxide level until individual colonies were visible to the naked eye (about 14-20 days). Clones were preliminarily screened using PCR for deletion of the intervening region between the two CRISPR cut sites, as described previously[63]. The genotype of the knock-out clones was confirmed by sequencing the genomic region encompassing the deleted sequence using the following primer pair: 5′-GGCCGCTGAAAGGAGAAG-3′ and 5′-GCCAAGCGGTACCAAGAT-3′. Abrogation of the protein encoded by the *C16orf72* gene in the knock-out clones was confirmed by western blot analysis.

To generate cells which express exogenous *C16orf72*, the open read frame (ORF) of this gene was cloned into a lentiviral plasmid which allowed for the expression of Flag/HA-tagged version of the protein. Briefly, ORF of *C16orf72* with flanking attB sites were PCR-amplified and the purified amplicon was cloned into pDONR223 donor vector using Gateway BP Clonase II Enzyme Mix (Invitrogen, #11789020) following manufacturer's recommendations. The ORF was then cloned into pLENTI-EF1a-N-Flag-HA-PURO destination vector using Gateway LR Clonase II Enzyme Mix (Invitrogen, #11791020). Next, lentivirus was produced by transfecting 293T cells with plasmids encoding for Flag/HA-C16orf72 and lentiviral components, Tat, VSV-G, Rev, Gag and Pol. U2OS wild-type cells and C16orf72 knock-out clones Δ.2 and Δ.3 were infected with the resulting lentivirus and subsequently selected with 2 µg/ml puromycin for at least one week. Expression of Flag/HA-tagged C16orf72 protein was confirmed by western blot analysis. As control, cells expressing Flag/HA peptides only (or herein referred to as Flag/HA-EV) were produced in parallel and this was generated by cloning the sequence 5′- TGACGCATCTAG-3′ (which contains two stop codons) in place of the *C16orf72* ORF.

### Extraction of whole-cell extracts and chromatin-enriched fractions
Cells were trypsinised and pelleted by spinning at 1000 x g for 5 min and then washed in PBS twice to completely remove growth medium. To obtain whole-cell extract, cell pellet was either directly dissolved in 1X SDS Loading Buffer (50 mM Tris-HCl pH 6.8, 2% SDS, 0.05% Bromophenol Blue, 10% glycerol, 0.1 M DTT) and boiled at 95–100 °C for 10 min; or lysed in Triton X-100 Lysis Buffer (100 mM NaCl, 1% Triton X-100, 50 mM Tris-HCl pH 8.0, 5 mM N-ethylmaleimide, 1 mM DTT, 1X protease inhibitor cocktail, 1 mM sodium fluoride, 10 mM β-glycerophosphate, 0.1 mM sodium orthovanadate) for 30 min at 4 °C, centrifuged at top speed for 30 min at 4 °C, supernatant isolated, mixed with 1X SDS Loading Buffer (50 mM Tris-HCl pH 6.8, 2% SDS, 0.05% Bromophenol Blue, 10% glycerol, 0.1 M DTT) and boiled at 95–100 °C for 10 min. To obtain chromatin-enriched fraction, cell pellet was lysed in Extraction Buffer 1 (10 mM Tris-HCl pH 7.5, 150 mM sodium chloride, 1.5 mM magnesium chloride, 0.1% Triton X-100, 0.34 M sucrose, 10% glycerol, 1 mM DTT, 1X protease inhibitor cocktail) for 10 min on ice, centrifuged at top speed for 5 min at 4 °C, supernatant removed, nuclear pellet lysed in Extraction Buffer 2 (3 mM EDTA, 0.2 mM EGTA, 1 mM DTT, 1X protease inhibitor cocktail) for 30 min on ice, centrifuged at 1700 x *g* for 5 min at 4 °C, supernatant removed completely, and remaining pellet dissolved in 1X SDS Loading Buffer (50 mM Tris-HCl pH 6.8, 2% SDS, 0.05% Bromophenol Blue, 10% glycerol, 0.1 M DTT) and boiled at 95–100 °C for 10 min.

### Immunofluorescence microscopy
$6–8 \times 10^4$ cells were seeded per coverslip, in the 24-well plates, and the next day treated with either 2 mM HU for 2 or 24 hours, or with 5 µM Olaprib for 3 hours. For Rad51 foci, cells were fixed with 3.7% paraformaldehyde (PFA) for 10 minutes at room temperature, or in case of S9.6, BRCA1 and Senataxin with 100% methanol for 15 minutes at −20 °C. After permeabilizing cells with 0.5% TritonX-100 (Sigma) in Phosphate buffered saline (PBS) for 10 minutes, cells were incubated in blocking buffer (5% BSA, 1% FBS in 0.5% TritonX-100 PBS for S9.6 staining, or 2% BSA in 0.5% TritonX-100 PBS for Rad51, BRCA1 and Senataxin) for 1 hour. For RNase H treatment in experiments that detected R-loops using S9.6, cells were fixed and permeabilized as above, prior to incubation with 5 units of RNase H (BioLabs M0297S NEB) for 120 minutes at 37 °C in a humidity chamber. Cells were incubated in primary antibody overnight at 4 °C in blocking buffer, followed by and 3 × 10-minute washes in PBS. Cells were incubated in secondary fluorescent antibody in blocking buffer for 1 hour at room temperature in the dark, followed by 3 × 10-minute washes in PBS. Coverslips were mounted on glass slides using VECTASHEILD mounting media with DAPI. Images were taken using IX71 Olympus confocal microscope using 60x and 100x objectives. All the images were analysed using ImageJ in the same manner.

### Ribonucleotide Excision Repair Alkaline assay
Cells were trypsinised and genomic DNA isolated using the QIAprep Spin Miniprep kit. DNA concentration was determined using a

NANODROP 2000 Spectrophotometer. 2.5 µg DNA was incubated with 0.3 N NaOH for 2.5 hours at 55°. Subsequently samples were loaded on alkaline agarose gels (0.7% Agarose, 50 mM NaOH, 1 mM EDTA, pH 8) and run for 18 hours at 1Volt/cm. Next, the gel was incubated for 2 hours in the neutralizing buffer (1 M Tris pH 7.6, 1.5 M NaCl). Gels were stained overnight with 0.5 M Ethidium Bromide and imaged under UV via Alpha Innotech Gel Imaging system. All the images were analysed using ImageJ in the same manner.

## Proximity ligation assay

$8 \times 10^4$ cells were seeded per coverslip, in the 24-well plates, and the next day treated with 2 mM HU for 2 hours before fixing with 100% methanol for 15 minutes at −20 °C. Cells on coverslips were then subjected to MERCK Duolink Proximity Ligation Assays following manufacturer's instructions (reagents used: DUO92004, DUO92002 and DUO92007). For negative controls, the protocol was performed with one primary antibody and omission of the second primary antibody. Fixed cells were incubated with the appropriate primary antibody combinations for 2 hours at 37 °C and then with PLUS and MINUS PLA probes for 70 minutes at 37 °C. Ligation and amplification steps were then performed for 30 and 100 minutes respectively, at 37 °C. Coverslips were fixed onto glass slides using the VECTASHEILD mounting media with DAPI. Images were taken using IX71 Olympus confocal microscope using 60x and 100x objectives. All the images were analysed using ImageJ in the same manner.

## Co-immunoprecipitation

Approximately $3 \times 10^7$ cells were used for each experiment. When the cells reached 60–80% confluence, they were washed twice with ice-cold 1X PBS and subsequently resuspended in Triton X-100 Lysis Buffer (100 mM NaCl, 1% Triton X-100, 50 mM Tris-HCl pH 8.0 supplemented with 5 mM N-ethylmaleimide, 1 mM DTT, 1X protease inhibitor cocktail, 1 mM sodium fluoride, 10 mM β-glycerophosphate, 0.1 mM sodium orthovanadate, 125 U/ml benzonase, 5 mM magnesium chloride) by scraping. Cell suspension was incubated at 4 °C for 30 min with constant agitation before cell debris was removed by spinning at top speed for 30 min. For immunoprecipitation of Flag-tagged proteins, anti-Flag M2 affinity gel (Sigma, A2220) was used at 4 °C for at least an hour. Cell lysate was added to the beads and mixed for at least 3 h at 4 °C. As negative control, equal volume of lysis buffer was added to the beads instead of the cell lysate. The beads were then washed five times with lysis buffer (minus the benzonase and magnesium chloride) and eluted by boiling in 2X SDS Loading Buffer (100 mM Tris-HCl pH 6.8, 4% SDS, 0.1% Bromophenol Blue, 20% glycerol, 0.2 M DTT) for 10 min.

## DNA fibre assay

DNA fibre analysis was carried out as described previously[64,65] with minor modifications. U-2-OS cells were incubated with 25 µM CldU for 20 minutes and pulsed with 250 µM IdU for a further 20 minutes. Where indicated, cells were exposed to 2 mM HU for 2 or 3 hours between additions of the two analogues. After labelling had finished, cells were harvested and DNA fibres spread onto microscope slides. Immunostaining of DNA was carried out as described previously[65]. The types of replication fork structure were quantified, and the lengths of labelled tracts were measured using ImageJ. Arbitrary lengths were converted into µm using scale bars captured on images using a Nikon E600 Eclipse equipped with a 60x oil lens or a IX71 Olympus confocal microscope using a 100x objective.

## Statistics

The relevant statistical tests are stated where applicable. In each case statistical significance was analysed: *$p < 0.05$; **$p < 0.01$; ***$p < 0.001$; ****$p < 0.0001$. Otherwise, analyses were classified as not significant (ns). Student's t tests were two-tailed and unpaired. All the statistical analysis was performed using GraphPad Prism 9.

## Reporting summary

Further information on research design is available in the Nature Portfolio Reporting Summary linked to this article.

## Data availability

All data generated or analysed during this study are included in this published article (and its supplementary information files), or available from the corresponding author on request. Source Data are provided with this paper. Material requests should be made to the corresponding author. Source data are provided with this paper.

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

## Acknowledgements

HeLa wild-type and RNASEH2A-null cells were a Gift from A Jackson (University of Edinburgh). This research was funded in whole, or in part, by the Wellcome Trust (Grant 102348/Z/13/Z) and MRC (Grants MR/P028284/1; MR/P018963/1; MR/V00896X/1; MR/W017350/1). For the purpose of Open Access, the author has applied a CC BY public copyright licence to any Author Accepted Manuscript version arising from this submission. JL was supported by AstraZeneca. JK was supported by a BBSRC iCASE Award with AstraZeneca (BB/R505948/1). MKR was supported by the University of Oxford EP Abraham Cephalosporin Fund. The IG-S laboratory is supported by a Cancer Research UK Career Development Fellowship (C62538/A24670).

## Author contributions

A.B.S. and M.K.R. performed all experiments with the exception of the screen (G.E.R., D.J., D.S. and D.E.), DNA fibre (Fig. 4; M.R.H.) and DDR marker analysis (Figs. 3 and 4a, e; J.K.). A.W. and N.S. contributed towards cell line generation. P.W., J.V.F. and I.G.-S. advised. M.K.R., A.B.S. and N.D.L. wrote the manuscript.

## Competing interests

The authors declare no competing interests.
