## [Peer Review File · Nature Communications]

C16orf72/HAPSTR1/TAPR1 functions with BRCA1/Senataxin to modulate replication-associated R-loops and confer resistance to PARP disruptionREVIEWER COMMENTS

Reviewer #1 (Remarks to the Author):

In this manuscript, Ramlee et al demonstrated novel function of C16orf72 in replication and R-loop resolution. The authors identified that loss of C16orf72 is synthetic lethal with PARP1/2 disruption from screening. In cells, loss of C16orf72 on one hand, slows progression of replication fork and increases level of stalled forks, on the other hand leads to R-loop accumulation after replication stresses in a PARP1/2 independent pathway. As the consequence, cells accumulate DNA damage and are sensitive to PARPi and ATRi. Mechanistically, the authors propose that C16orf72 could promote BRCA/Senetaxin for R-loop resolution upon replication stress. Overall, the major findings in this manuscript are interesting and novel because function of C16orf72 in replication and R-loop regulation was not reported in previous studies. The paper is clearly written and easy to follow up. However, some critical questions are unanswered and additional evidence is required in this study for explaining the mechanisms. Major conclusions are based on imaging assays, which are not convincing. Technically, quantification of staining data needs to be clarified and improved.

Major issues

1. In the absence of exogenous replication stress, loss of C16orf72 already leads to a significantly reduced fork progression and increased stalled forks (Fig. 4b and Fig. 4c). It is not clear how does C16orf72 affect replication progression. Is C16orf72 essential for replication both before and after stress? Does C16orf72 knock out (KO) affect proliferation rate without exogenous replication stress? Is C16orf72 localized at replication forks in cells or bind to replication fork in vitro? Efforts to answer some basic function of C16orf72 in replication are expected.
2. In contrast to the effect on replication, C16orf72 KO does not change level of R-loops without HU (Fig. 5d). R-loops are accumulated only after HU. In the absence HU treatment, is replication-stress-induced R-loops resolved by PARP-dependent pathway? Would PARP1/2 depletion in C16orf72 KO increase level of R-loops without additional replication stress? It would be helpful to comment whether C16orf72-dependent R-loop suppression is triggered only in case of replication stress overcomes the threshold.
3. In Fig. 2b, the authors count intensity/nucleus of gammaH2AX. Representative images for control group and HU treated group are shown with different numbers of cells. gammaH2AX intensity in each cell varies in a large range due to endogenous stresses and cell cycle states. For understanding global change of bulk cell population, it is fair to quantify percentage of gammaH2AX foci positive cells rather than intensity/nucleus.
4. The same is for RAD51 foci in Sup Fig. 5; BRCA1 foci, and Senetaxin foci numbers in Fig. 6. How does a cell look like with 100 or 300 foci/nucleus in Fig. 6? In untreated cells, BRCA1 does not form foci in many cells, which are reflected in the quantification data. In addition to foci numbers and/or intensity, percentage of foci positive cells needs to be quantified.
5. S9.6 mediated R-loop staining in Fig. 5 is problematic. Again, there are no representative images of S9.6 staining anywhere in the figures. R-loop accumulation could be restricted to transcriptionally active sites. Does the signal increase of S9.6 refer to homogenous or local increase (foci like staining) in the nucleus?
6. As described in No.2, BRCA1/Senetaxin foci quantification are questionable. Therefore, change of BRCA1/Senetaxin foci after HU in cells with or without C16orf72 is not convincing to explain that C16orf72 promote BRCA-senetaxin mediated R-loop resolution.

Minor issues

1. Supplementary 3 is less relevant to this study.
2. Line 193 and Line 1046: Spelling of "DNA fiber assay" is rather than "fibre"
3. Effect of siRNaseH2A is not quite effective in WB (Sup 6c).

Reviewer #2 (Remarks to the Author):

This manuscript provides important and novel insight into the role of c16orf72 on the modulation of replication-associated R-loops acting in the same pathway as BRCA1. The authors use a CRISPR screen to identify genes that are synthetic lethal with PARP1 and PARP2 and identify c16orf72 among others. After validating the synthetic lethality between PARP1 and c16orf72, they systematically analyze the function of c16orf72. They build on previous studies that have shown c16orf72 to function in the replication stress response and find that c16orf72 is important for replication fork recovery. After probing different pathways underlying PARP inhibitor resistance in search for the function of c16orf72, they narrow down to R-loop formation and modulation. They identify that c16orf72 work together with BRCA1 and Senataxin and facilitates the assembly of BRCA1 and Senataxin at sites of replication stress. The manuscript is well written, the figures are good quality and easy to interpret, the findings are important, the conclusions are supported by the presented data. It should be published after minor revisions.

Minor points

1. HUWE1 is a poly(ADP-ribose)-binding protein (Wang et al, Genes&Dev 2012) and was shown that its depletion reverses PARP inhibitor sensitivity in BRCA2 deficient cells (Clements et al, Nat Comm 2020). Therefore the interaction between c6orf72 and HUWE1 is very interesting and would deserve a more detailed analysis the current paragraph (only two sentences) and data in supplementary figures only. I would consider excluding it from this manuscript since the way it is written it is not critical.
2. Maybe I misunderstand this sentence: "Strikingly, whilst depletion of DDX5, ATAD5 or DHX9 did not further elevate R-loops in response to HU (Supplementary Data 8), this did occur upon depletion of BRCA1 (Figure 6c)." When looking at the figures, it appears to me that BRCA1 depletion is similar to the depletion of DDX5, ATAD5 or DHX9: there does not seem to be a further increase in R-loops in response to HU.
3. On Figure 3b, gH2AX is convincing upon HU treatment but maybe a few more cells could have been shown in the micrographs. Is magnification the same? The sinnlge nuclei appear larger. There is no scale bar.
4. On figure 4e, on the U2OS Aphidicolin negative merged panel Cyclin A is not readily visible, also maybe the images are cropped with a bit of shift between channels. Also, the c16orf72Δ.1 Aphidicolin negative merged panel nuclei appear to be smaller than on the individual channels.
5. On figure 5b, the PLA signal is great and convincing. It'd be great to see more nuclei in the image to see PLA in more instances.

Reviewer #3 (Remarks to the Author):

In this report, the authors used CRISPR/cas9 to KO PARP1, PARP2 and both PARP1&PARP2 in U2OS cells to then run genome wide gRNA screen covering 18,053 genes. The premise is to discover "novel genes that suppress toxicity of PARP1/2 gene disruption". The major flaw here is that there is no toxicity due to PARP1/2 gene disruption. It is noted however than several groups have reported the development of U2OS/PARP1-KO, U2OS/PARP2 and U2OS/PARP1-KO/PARP2-KO cells, with no reported toxicity.

In this regard, the basic premise is a bit flawed and so it is surprising that they seem to have discovered a novel gene product C16orf72 that works with BRCA1/Senataxin in the resolution of R-loops. It may have been more effective had they screened for genes that suppress toxicity of PARP1/2 gene disruption after BRCA1-shRNA, for example.

However, in support of the claim, they do show that loss of C16orf72 sensitizes cells to loss or suppression of PARP1/2 (Figure 1B,1C) and sensitizes cells to PARPi (Figure 1D) and ATRi (Figure 1E).

These data and that in Figure 2 nicely support a role for C16orf72 in replication stress, although the source was not clear.

Some major concerns include:

The data in Figure 3A is far from convincing, with only minor changes in the gH2AX, pRPA and p-Chk1 levels in response to HU and mostly in only one KO only. This is concerning and suggests a need for more detailed analysis and quantification.

The plot for Figure 3B only shows 3 data points per bar – is that three cells only? This analysis seems flawed. At least 50 cells should be analyzed and pan gH2AX staining is not normally representative of replication stress, as opposed to gH2AX foci.

Similarly, the plot in Figure 3C should show all data points.

Similarly, the plot in Figure 4A should show all data points. At least 50 cells should be analyzed and pan gH2AX staining is not normally representative of replication stress, as opposed to gH2AX foci.

Figure 5C,D should include the complemented cell line.

A key claim here is that C16orf72 forms a complex with BRCA1 and Senataxin, as suggested by the data shown in Figure 5E. However, lacking is the standard reverse IP analysis (after pull-down of BRCA 1 and after pull-down of Senataxin). This should be included.

Many of the plots in Figures 1, 2, 3, 4 and 7 only show the averages – all data points should be plotted.

NCOMMS-23-04549 Response to Reviewer's Comments

Reviewer #1

In this manuscript, Ramlee et al demonstrated novel function of C16orf72 in replication and R-loop resolution. The authors identified that loss of C16orf72 is synthetic lethal with PARP1/2 disruption from screening. In cells, loss of C16orf72 on one hand, slows progression of replication fork and increases level of stalled forks, on the other hand leads to R-loop accumulation after replication stresses in a PARP1/2 independent pathway. As the consequence, cells accumulate DNA damage and are sensitive to PARPi and ATRi. Mechanistically, the authors propose that C16orf72 could promote BRCA/Senetaxin for R-loop resolution upon replication stress. Overall, the major findings in this manuscript are interesting and novel because function of C16orf72 in replication and R-loop regulation was not reported in previous studies. The paper is clearly written and easy to follow up. However, some critical questions are unanswered and additional evidence is required in this study for explaining the mechanisms. Major conclusions are based on imaging assays, which are not convincing. Technically, quantification of staining data needs to be clarified and improved.

We thank this reviewer for their constructive comments and for considering the major findings in this manuscript are interesting and novel and that the paper is clearly written and easy to follow. Several specific comments were raised, particularly with regards technical aspects of the imaging data. We have included significant new analysis to address these comments that we outline below:

Major issues

1. In the absence of exogenous replication stress, loss of C16orf72 already leads to a significantly reduced fork progression and increased stalled forks (Fig. 4b and Fig. 4c). It is not clear how does C16orf72 affect replication progression. Is C16orf72 essential for replication both before and after stress? Does C16orf72 knock out (KO) affect proliferation rate without exogenous replication stress? Is C16orf72 localized at replication forks in cells or bind to replication fork in vitro? Efforts to answer some basic function of C16orf72 in replication are expected.

As pointed out by this reviewer, it is clear that loss of C16orf72 impacts on both unperturbed replication fork speeds and recovery/restart of forks, even in the absence of exogenous replication stress induced by HU (Figure 4b, c). As suggested, it is tempting to speculate that C16orf72 may function during unperturbed replication, perhaps as a component of the replication fork. As requested, we have assessed the proliferation rate of *c16orf72* Δ cells.

This reveals that whilst *c16orf72*Δ cells show a subtle but insignificant reduction in proliferation rate relative to parental control cells, this is not dependent of the expression of C16orf72 (Supplementary Figure 5a). There is also no significant change in the cell cycle profiles of *c16orf72*Δ cells relative to *c16orf72*Δ expressing recombinant Flag-C16orf72 (Supplementary Figure 5b). Additionally, the absence of C16orf72 in chromatin fractions in the absence of HU (Figure 2d) and published iPOND mass spectrometry datasets of unstressed cells (Sirbu et al., [2012] Nat. Protoc., 7:594; and Dungrawala et al., [2015], Mol. Cell, 59:998) suggest it is not an intrinsic component of the replisome. Consistent with this, in contrast to PCNA we do not see co-localisation of C16orf72 with the replicative helicase (MCM2) in proximity ligation assays (Supplementary Figure 5c). Together, these data suggest C16orf72 is not a component of the replication fork. Instead, we feel it is more likely that the increased fork stalling/decreased fork speeds observed in *c16orf72*Δ cells in the absence of HU is due to ‘endogenous’ replication stress that occurs during DNA synthesis, perhaps at specific loci/areas of the genome that are difficult to replicate. We have included these data (Supplementary Figure 5) and discussed these possibilities in the results (p7).

2. In contrast to the effect on replication, C16orf72 KO does not change level of R-loops without HU(Fig. 5d). R-loops are accumulated only after HU. In the absence HU treatment, is replication-stress-induced R-loops resolved by PARP-dependent pathway? Would PARP1/2 depletion in C16orf72 KO increase level of R-loops without additional replication stress? It would be helpful to comment whether C16orf72-dependent R-loop suppression is triggered only in case of replication stress overcomes the threshold.

We included data in the original manuscript illustrating PARPi induce R-loops in the absence of HU, indicating a role for PARP1/2 in suppressing R-loops in the absence of exogenously applied replication stress (Figure 7d). As suggested by this reviewer, this experiment also included data demonstrating that disruption of *C16orf72* further elevates R-loops induced by PARPi. As eluded to by this reviewer, these data indicate C16orf72 regulates R-loop homeostasis in conditions other than HU-induced replication stress, specifically under conditions of PARP1/2 disruption. As requested, we have commented on this in the results (p11) and discussion (p15).

3. In Fig. 2b, the authors count intensity/nucleus of gammaH2AX. Representative images for control group and HU treated group are shown with different numbers of cells. gammaH2AX intensity in each cell varies in a large range due to endogenous stresses and cell cycle states. For understanding global change of bulk cell population, it is fair to quantify percentage of gammaH2AX foci positive cells rather than intensity/nucleus.

As requested, we now express the data as the % of γ H2AX positive cells in Figure 3a. For consistency we have also performed this analysis for the γ H2AX data presented in Figure 4a. In both cases this does not change the conclusions drawn from the data. The data expressed as mean γ H2AX nuclear intensity have now been moved to Supplementary Figure 4a and b illustrating all data points, as requested by reviewer #3.

4. The same is for RAD51 foci in Sup Fig. 5; BRCA1 foci, and Senetaxin foci numbers in Fig. 6. How does a cell look like with 100 or 300 foci/nucleus in Fig. 6? In untreated cells, BRCA1 does not form foci in many cells, which are reflected in the quantification data. In addition to foci numbers and/or intensity, percentage of foci positive cells needs to be quantified.

As requested, we represent the % cells with Rad51 foci in Supplementary Figure 7a. These data clearly show *c16orf72* Δ cells are competent for Rad51 nuclear foci in response to replication stress, particularly at early time points (referred to on p7). For Senataxin and BRCA1 foci, consistent with a previous analysis (Yüce *et al.*, 2013. *Molecular and Cellular Biology*, 33:406) we have analysed the distribution of cells with different foci numbers, including representative images of foci (Supplementary Figure 11b and c). These data illustrate a reduction in the % of *c16orf72* Δ cells that exhibit >100 or >30 Senataxin and BRCA1 foci respectively after HU exposure, supporting our conclusions that C16orf72 is required for optimal assembly of these proteins into chromatin following replication stress.

5. S9.6 mediated R-loop staining in Fig. 5 is problematic. Again, there are no representative images of S9.6 staining anywhere in the figures. R-loop accumulation could be restricted to transcriptionally active sites. Does the signal increase of S9.6 refer to homogenous or local increase (foci like staining) in the nucleus?

We illustrated representative images of S9.6 staining in Supplementary Figure 7a of the original manuscript (now Supplementary Figure 9a). As indicated in this figure, quantification in the original manuscript was total nuclear S9.6 staining. Within nuclei we do observe focal S9.6 staining. Therefore, we have also quantified these structures and included the data in Supplementary Figure 9c and referred to this data in the text (p8-9). This analysis also indicates elevated R-loops in *c16orf72* Δ cells and so does not change our overall conclusions.

6. As described in No.2, BRCA1/Senetaxin foci quantification are questionable. Therefore, change of BRCA1/Senetaxin foci after HU in cells with or without C16orf72 is not convincing

to explain that C16orf72 promote BRCA-senetaxin mediated R-loop resolution.

Please see response to point 4.

Minor issues

1. Supplementary 3 is less relevant to this study.

Following this comment, in addition to the suggestion of Reviewer #2 (point 1), we have removed Supplementary Figure 3 and the associated text (p5, 12/13, 16, 21 and 22) from the revised manuscript.

2. Line 193 and Line 1046: Spelling of “DNA fiber assay” is rather than “fibre”.

This has been corrected throughout the manuscript.

3. Effect of siRNAseH2A is not quite effective in WB (Sup 6c).

Whilst residual RNaseH2A is evident in the siRNA knockdown (Supplementary Figure 8c) it is important to note that this phenocopies the *RNASEH2A* knockout cells with respect to R-loop homeostasis – specifically, showing increased R-loop accumulation in response to HU relative to control cells (compare Fig. 6a and b). We therefore believe this level of knockdown is sufficient for the epistasis analysis that indicates C16orf72 and RNaseH2A do not function in the same pathway to suppress HU-induced R-loop accumulation.

Reviewer #2

This manuscript provides important and novel insight into the role of c16orf72 on the modulation of replication-associated R-loops acting in the same pathway as BRCA1. The authors use a CRISPR screen to identify genes that are synthetic lethal with PARP1 and PARP2 and identify c16orf72 among others. After validating the synthetic lethality between PARP1 and c16orf72, they systematically analyze the function of c16orf72. They build on previous studies that have shown c16orf72 to function in the replication stress response and find that c16orf72 is important for replication fork recovery. After probing different pathways underlying PARP inhibitor resistance in search for the function of c16orf72, they narrow down to R-loop formation and modulation. They identify that c16orf72 work together with BRCA1 and Senataxin and facilitates the assembly of BRCA1 and Senataxin at sites of replication stress. The manuscript is well written, the figures are good quality and easy to

interpret, the findings are important, the conclusions are supported by the presented data. It should be published after minor revisions.

We thank this reviewer for their positive comments. We have addressed their comments in the revised manuscript as outlined below:

Minor points

1. HUWE1 is a poly(ADP-ribose)-binding protein (Wang et al, Genes&Dev 2012) and was shown that its depletion reverses PARP inhibitor sensitivity in BRCA2 deficient cells (Clements et al, Nat Comm 2020). Therefore the interaction between c6orf72 and HUWE1 is very interesting and would deserve a more detailed analysis the current paragraph (only two sentences) and data in supplementary figures only. I would consider excluding it from this manuscript since the way it is written it is not critical.

In hindsight we can see that our data concerning the link between HUWE1 and C16orf72 is peripheral to this study. Given this point was also raised by Reviewer 1 (minor issues; point 1), we have removed this data (Supplementary Figure 3) and associated text (p5, 12/13, 16, 21 and 22) from the revised manuscript, as suggested.

2. Maybe I misunderstand this sentence: “Strikingly, whilst depletion of DDX5, ATAD5 or DHX9 did not further elevate R-loops in response to HU (Supplementary Data 8), this did occur upon depletion of BRCA1 (Figure 6c).” When looking at the figures, it appears to me that BRCA1 depletion is similar to the depletion of DDX5, ATAD5 or DHX9: there does not seem to be a further increase in R-loops in response to HU.

We thank the reviewer for pointing this out. The error for this is on our part due to our re-arranging figure panels and unintentionally misrepresenting the workflow as if we initiated these studies by studying BRCA1 as opposed to Senataxin. How our experiments evolved was that we initially tested whether various factors that had previously been implicated in R-loop homeostasis (Senataxin, DHX9, DDX5 and ATAD5) suppressed HU-induced R-loops, similar to C16orf72. Of these factors the only one that replicates C16orf72 for this phenotype is Senataxin. We subsequently identified that Senataxin and c16orf72 are epistatic with regards to this phenotype, indicating they function in the same pathway to suppress replication stress-induced R-loops. Given the link between Senataxin and BRCA1 in R-loop homeostasis, we next assessed the link between BRCA1 and C16orf72. We have re-written this aspect of the manuscript to illustrate this workflow (p9-10).

3. On Figure 3b, γ H2AX is convincing upon HU treatment but maybe a few more cells could have been shown in the micrographs. Is magnification the same? The single nuclei appear larger. There is no scale bar.

We have amended the micrographs as requested to include more cells with the addition of a scale bar. Please note this is now Figure 3a.

4. On figure 4e, on the U2OS Aphidicolin negative merged panel Cyclin A is not readily visible, also maybe the images are cropped with a bit of shift between channels. Also, the *c16orf72* Δ .1 Aphidicolin negative merged panel nuclei appear to be smaller than on the individual channels.

We have modified the images in Figure 4e to clarify cyclin A staining and highlight 53BP1 bodies in cyclin A-negative cells. We have also rectified the cropping and include scale bars on the images.

5. On figure 5b, the PLA signal is great and convincing. It'd be great to see more nuclei in the image to see PLA in more instances.

As requested, we have included more nuclei in the images for Figure 5b.

Reviewer #3

In this report, the authors used CRISPR/cas9 to KO PARP1, PARP2 and both PARP1&PARP2 in U2OS cells to then run genome wide gRNA screen covering 18,053 genes. The premise is to discover "novel genes that suppress toxicity of PARP1/2 gene disruption". The major flaw here is that there is no toxicity due to PARP1/2 gene disruption. It is noted however that several groups have reported the development of U2OS/PARP1-KO, U2OS/PARP2 and U2OS/PARP1-KO/PARP2-KO cells, with no reported toxicity.

In this regard, the basic premise is a bit flawed and so it is surprising that they seem to have discovered a novel gene product *C16orf72* that works with BRCA1/Senataxin in the resolution of R-loops. It may have been more effective had they screened for genes that suppress toxicity of PARP1/2 gene disruption after BRCA1-shRNA, for example.

However, in support of the claim, they do show that loss of *C16orf72* sensitizes cells to loss

or suppression of PARP1/2 (Figure 1B,1C) and sensitizes cells to PARPi (Figure 1D) and ATRi (Figure 1E). These data and that in Figure 2 nicely support a role for C16orf72 in replication stress, although the source was not clear.

We thank this reviewer for their comments and considering we perform experiments that illustrate loss of C16orf72 sensitizes cells to PARP1/2 disruption either genetically or through PARPi, and that nicely support a role for C16orf72 in the cellular response to replication stress.

However, there seems to have been some confusion about the initial goal that underpins this manuscript, particularly with regards to the screen that we performed to initiate this study. This confusion has arisen through what we mean by identifying '*genes that suppress toxicity of PARP1/2 gene disruption*'. As pointed out by this reviewer, *parp1parp2* knock-out cells are viable. Therefore, our aim was to screen for genes that when disrupted lead to reduced viability of *parp1parp2* knock-out cells - i.e. genes that are required to maintain the viability of *parp1parp2* Δ cells (or suppress toxicity) - in other words genes that are synthetic lethal with PARP1/PARP2. In hindsight we can see that the wording of our original statement is over-elaborate and could lead to confusion. We have therefore re-phrased this in the abstract (*...we conducted a genome-wide screen for genes that are synthetic lethal with PARP1/2 gene disruption*) or in the Introduction (*'we identify C16orf72/HAPSTR1/TAPR1 as a gene that is synthetic lethal with PARP1/2 gene disruption and allows cells to tolerate replication stress'*; p4).

The reviewer suggests an alternative screen to identify genes that when mutated lead to resistance of BRCA1-depleted cells to *PARP1/2* gene disruption. This is worthwhile with the potential to identify genes whose loss of function leads to BRCA-defective cells becoming resistant to PARP inhibitors. However, this is distinct from the initial goal of this study which was to identify novel genes that are synthetic lethal with *PARP1/2* gene disruption and as such is beyond the scope of the current report.

Some major concerns include:

1. *The data in Figure 3A is far from convincing, with only minor changes in the gH2AX, pRPA and p-ChK1 levels in response to HU and mostly in only one KO only. This is concerning and suggests a need for more detailed analysis and quantification.*

Whilst these observations are reproducible we do not have sufficient repeats of certain blots to quantify the data with supporting statistical analysis. Given we indicate elevated DNA

damage and replication difficulties in *c16orf72* knockout cells though a variety of other approaches (Elevated and persistent γ H2AX, defective replication dynamics, elevated 53BP1 bodies) we have omitted these data from the revised manuscript. This does not change the workflow, conclusions, or overall take-home message of the manuscript. The text has been revised accordingly (p6).

2. The plot for Figure 3B only shows 3 data points per bar – is that three cells only? This analysis seems flawed. At least 50 cells should be analyzed and pan γ H2AX staining is not normally representative of replication stress, as opposed to γ H2AX foci.

The data illustrated in the original Figure 3b represented the mean nuclear intensity of γ H2AX derived from 3 independent experiments. As stated in the figure legend of the original manuscript, the values are derived from at least 500 cells in each biological repeat. The 3 individual data points shown on the graph illustrate the mean nuclear intensity of γ H2AX values for each of the 3 biological repeats, not individual cells.

There are several ways to represent these data, including plotting each individual value, as requested by this reviewer, or representing the % cells with γ H2AX, as requested by reviewer 1. The data illustrated in the original Figure 3b and 4a are high content imaging data based on the quantification of several thousand cells (see source data file). As such, plotting each individual time point results in so many individual data points that it becomes visually saturated, making it difficult to see any differences within datasets. It is for this reason we chose to represent mean nuclear intensity in the original figure. Given reviewer #1 asked us to represent data as the % γ H2AX-positive cells, we would prefer to use this analysis as the trends are more clearly seen. We have therefore included this analysis for Figures 3b (now Figure 3a) and 4a. Having said that, we understand the importance of plotting each individual data point, including an analysis of mean nuclear γ H2AX intensity. Therefore, as requested we have included this data in Supplementary Figure 4b and c.

3. Similarly, the plot in Figure 3C should show all data points.

All data points are represented in the left-hand panels of this figure, as in the original submission. Please note this figure is now Figure 3b.

4. Similarly, the plot in Figure 4A should show all data points. At least 50 cells should be analyzed and pan γ H2AX staining is not normally representative of replication stress, as

opposed to γ H2AX foci.

See response to point 2.

5. Figure 5C,D should include the complemented cell line.

Figure 5c is PLA data performed with different antibody combinations in parental U2OS cells that contain endogenous c16orf72. Therefore, complementation is not possible for this experiment as we are not using a c16orf72 knock-out cell line.

With regards to the c16orf72 knockout cell lines employed in Figure 5d, the complementation of clone c16orf72 Δ .2 is illustrated in the original Figure 5e. The equivalent experiment for clone c16orf72 Δ .3 was represented in Supplementary Figure 7b of the original submission (now Supplementary Figure 9b).

6. A key claim here is that C16orf72 forms a complex with BRCA1 and Senataxin, as suggested by the data shown in Figure 5E. However, lacking is the standard reverse IP analysis (after pull-down of BRCA 1 and after pull-down of Senataxin). This should be included.

Unfortunately, we have been unable to verify successful IP of BRCA1 due to limitations in the availability of antibody combinations that can detect BRCA1 by western blotting following a BRCA1 IP. Therefore, given the previously established interaction between BRCA1 and Senataxin (Hatchi E, *et al. Molecular cell* **57**, 636 [2015]; *Nature communications* **8**, 15908 [2017]) we focussed our attention of performing the requested reverse IP with Senataxin. In this context we can IP Senataxin from extracts and these experiments have successfully identified co-IP of C16orf72, verifying the Senataxin-C16orf72 interaction through a reverse IP (Supplementary Figure 11a and p10).

7. Many of the plots in Figures 1, 2, 3, 4 and 7 only show the averages – all data points should be plotted.

We now include plots that include all data points, specifically for Figure 3 (see response to points 2 and 3) and Figure 4a (see response to points 4). We have also performed this analysis in Figure 4e (Supplementary Figure 6). This does not alter the statistical significance of any changes highlighted and as such the conclusions we draw from the data.

However, in certain data sets highlighted by this reviewer this type of analysis is not possible. In these instances we have represented data using standard procedures for the field. For example, data in Figures 1 and 2 are clonogenic survival assays. In these assays it is usual to represent data as the % survival of treated cells relative to untreated controls. Therefore, it is not possible to plot individual values. The same applies for the clonogenic survival assays represented in figures 7b, c, e and f. For DNA fiber analysis represented in figures 4c, d and 7a, the number of specific DNA structures are scored. It is standard practice to represent these data as % of stalled forks or new origins.

REVIEWERS' COMMENTS

Reviewer #1 (Remarks to the Author):

I have read the revised manuscripts. Additional experiments provided sufficient evidence to support their conclusion. I recommend publication of revised manuscript.

Reviewer #2 (Remarks to the Author):

The authors addressed my concerns and much improved the manuscript. I suggest accepting it for publication.

Reviewer #3 (Remarks to the Author):

The authors have very effectively addressed the concerns from the previous review. The manuscript is now very logically worded, and the data strongly supports the overall hypothesis proposed and conclusions drawn.